# Learning Individual Behavior in Agent-Based Models with Graph Diffusion Networks

**Francesco Cozzi** ⬥
Sapienza University, Rome, Italy
CENTAI, Turin, Italy
`f.cozzi@uniroma1.it`

**Marco Pangallo** ⬥
CENTAI, Turin, Italy
`marco.pangallo@centai.eu`

**Alan Perotti** ⬥
CENTAI, Turin, Italy
`alan.perotti@centai.eu`

**André Panisson** ⬥
CENTAI, Turin, Italy
`panisson@centai.eu`

**Corrado Monti** ⬥
CENTAI, Turin, Italy
`me@corradomonti.com`

## Abstract

Agent-Based Models (ABMs) are powerful tools for studying emergent properties in complex systems. In ABMs, agent behaviors are governed by local interactions and stochastic rules. However, these rules are, in general, non-differentiable, which limits the use of gradient-based methods for optimization, and thus integration with real-world data. We propose a novel framework to learn a differentiable surrogate of any ABM by observing its generated data. Our method combines diffusion models to capture behavioral stochasticity and graph neural networks to model agent interactions. Distinct from prior surrogate approaches, our method introduces a fundamental shift: rather than approximating system-level outputs, it models individual agent behavior directly, preserving the decentralized, bottom-up dynamics that define ABMs. We validate our approach on two ABMs (Schelling's segregation model and a Predator-Prey ecosystem) showing that it replicates individual-level patterns and accurately forecasts emergent dynamics beyond training. Our results demonstrate the potential of combining diffusion models and graph learning for data-driven ABM simulation.

## 1 Introduction

Agent-Based Models (ABMs) are computational frameworks in which autonomous "agents" interact with each other and their environment, leading to emergent collective behavior [43]. ABMs are typically characterized by: (i) a well-defined network of interactions, where the state of each agent is influenced by the states of a specific set of other agents, usually from the previous time step; (ii) stochasticity, meaning that agents' decisions incorporate a degree of randomness, producing probability distributions over multiple runs that capture real-world uncertainty and variation. ABMs have proven to be a powerful tool for developing and refining theoretical understanding, particularly in identifying minimal sets of micro-level rules that generate realistic macro-level outcomes [34]. In this sense, they have been instrumental in modeling a diverse range of phenomena [6], including structure formation in biological systems, pedestrian traffic, urban aggregation, and opinion dynamics. More recently, ABMs have demonstrated their value as forecasting tools [32], such as in predicting the economic impacts of the COVID-19 pandemic [30].

However, this progress is occurring despite the absence of principled methods to systematically align ABMs with real-world data. While various approaches have been proposed for calibrating macro-level parameters of ABMs [31], there are still no established methods for tuning the micro-level behaviors

39th Conference on Neural Information Processing Systems (NeurIPS 2025).

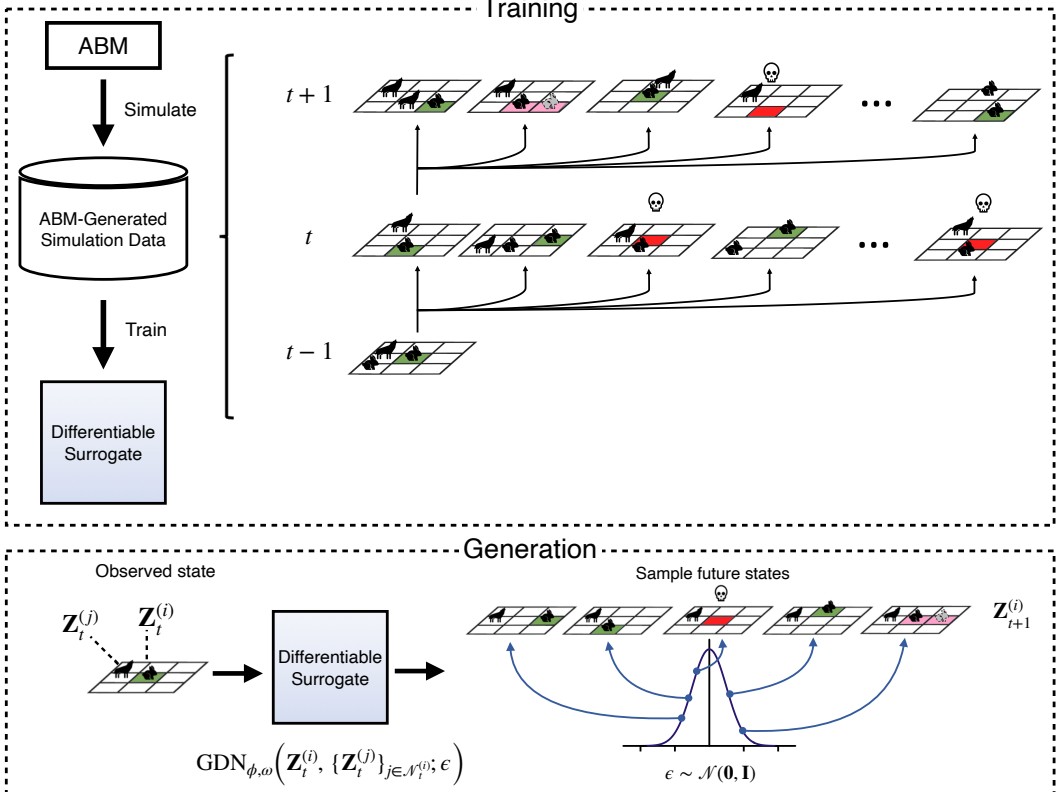

Figure 1: Overview of the training and generation pipeline for differentiable surrogates of Agent-Based Models. The top-left panel illustrates the training process: we run simulations using the original ABM, and use the resulting trajectories to train the differentiable surrogate. The top-right panel shows the structure of the ABM-generated data using the Predator-Prey model as an example: a state at time step $t-1$ gives rise to multiple possible states at time $t$, one of which is chosen to generate further possible states at $t+1$. Colored cells highlight the behavior of a specific "prey" agent — green for "move," red for "die," and pink for "reproduce." The bottom panel shows the generation phase: given a new observed state, the trained surrogate simulates plausible future states.

of individual agents to match observed data. One potential approach is to manually construct a probabilistic model that replicates the ABM and then use its likelihood function to estimate individual state variables [25]. However, this method requires the manual development of an ad-hoc probabilistic framework that reproduces the original ABM. Thus, what is currently missing is a fully automated method for deriving a learnable, differentiable model directly from an ABM.

In this work, we propose a novel approach to address this challenge: combining a graph neural network with a diffusion model to learn a differentiable *surrogate* of an ABM, from ABM-generated data. We refer to this method as a *Graph Diffusion Network (GDN)*. In this framework, a graph neural network captures the interactions that govern the evolution of each agent's state in response to other agents, while the diffusion model learns the distribution of possible state transitions, conditioned on these interactions. A central aspect of our approach is its explicit modeling of individual agent behavior. Rather than treating the system as a whole, we focus on how each agent acts as an independent entity, while also incorporating the influence of other agents on its decisions. This approach ensures that the emergent dynamics remain faithful to the decentralized nature of ABMs. By constraining the surrogate to model only micro-level rules, it cannot rely on shortcuts that predict only macro-level outcomes, preserving the distributed logic of the original ABM.

Our approach draws inspiration from previous work on using neural network models to emulate deterministic cellular automata [13]. However, we extend this idea to the broader domain of ABMs by introducing a crucial component: *stochasticity*. By incorporating stochasticity, our architecture can learn directly from ABM-generated data traces, making it adaptable to a wide variety of agent-based

models across diverse real-world applications. Furthermore, since our method is trained on data traces, it can seamlessly integrate empirical observations alongside simulated data, thus being potentially applicable to real-world scenarios. In this sense, our work represents a first step toward developing a comprehensive methodology for creating easy-to-use, learnable ABMs.

## 2   Background

From a general perspective, an ABM can be represented as a stochastic process $\mathbf{Z}_t \sim \mathbb{P}_\Theta(\mathbf{Z}_t \mid \mathbf{Z}_{\tau < t})$, where $\mathbf{Z}_t$ denotes the *state variables* at time $t$, $\Theta$ is a set of *parameters*, and $\mathbb{P}$ is a probability distribution implicitly defined by the model structure and parameters. The index $t$ represents discrete time. Typically, $\Theta$ consists of a small number of parameters, remains fixed in dimension, is interpretable by domain experts, and serves as the model's primary control mechanism. Conversely, each element in $\mathbf{Z}_t$ captures an agent's state, leading to a high-dimensional state space.

To illustrate this structure, we consider two ABMs used throughout the paper. The first is the well-known model by Schelling [36], where $\mathbf{Z}_t$ captures agents' positions and *colors*, and $\Theta$ indicates preference for same-color neighbors. Even with some tolerance for neighbors of different colors, agents often form segregated clusters [41]. This clear mismatch between individual preferences and aggregate outcomes is a classic example of *emergence*. The second model is a predator-prey model [39, 43], describing the ecological dynamics between two interacting species, with one acting as predator and the other as prey, similarly to the Lotka-Volterra equations. In this ABM, $\mathbf{Z}_t$ includes agent position and type (prey-predator), while $\Theta$ governs the probability to move, reproduce, or die. This model replicates the cyclical predator-prey population dynamics, typical of Lotka-Volterra systems, while also capturing complex spatial patterns reminiscent of spatial evolutionary games [27]. Both the Schelling and predator-prey models are widely recognized as canonical ABMs and are frequently used as testbeds for the development of novel calibration and surrogate techniques [37, 16, 23, 39, 40, 26].

ABMs have traditionally been powerful for theory generation, but in recent years, they have become increasingly data-driven [29]. To align ABM output with empirical data, most efforts focus on calibrating parameters $\Theta$ so that model-generated summary statistics match observed ones [31, 33]. Less attention, however, has been paid to estimating agent states $\mathbf{Z}_t$, which is key for matching time series further to summary statistics. Some researchers use data assimilation methods like particle filters [22] or ensemble Kalman filters [28] to infer $\mathbf{Z}_t$. A more principled alternative is to make ABMs differentiable, enabling the maximization of a likelihood function via gradient descent and automatic differentiation [24, 25]. While differentiability is straightforward for simple stochastic behaviors, such as those governed by Bernoulli trials [3], it becomes far more challenging for complex behaviors like those observed in the Schelling and predator-prey models.

To address this and other challenges in ABMs, researchers have increasingly turned to more tractable surrogates, also known as meta-models or emulators [17, 9, 11]. Surrogate models typically learn directly the mapping from parameters $\Theta$ to static summary statistics, disregarding individual behavior and model dynamics. For instance, a surrogate in Lamperti et al. [20] maps $\Theta$ to the mean growth rate of the economy. More recent research has also explored the emulation of model dynamics. Grattarola et al. [13] use Graph Neural Networks to approximate cellular automata, which can be seen as a special case of ABMs with deterministic interaction rules. Dyer et al. [8] propose Ordinary Differential Equation emulators to construct interventionally consistent surrogates, ensuring that micro-state interventions produce results aligned with macro-state interventions. Casert et al. [5] employ Transformers to model the transition of physical systems from one *configuration* to another, in terms of their transition rates rather than reproducing individual agent behavior. Their method is tailored to physical systems in continuous time, making it inapplicable for interacting agents in the general case, since it requires explicitly enumerating allowed transitions between configurations.

In contrast to these approaches, our work is the first to jointly emulate *individual* and *stochastic* interacting agents. This is particularly important, since ABMs are inherently stochastic and rely on individual-level interactions to produce emergent aggregate outcomes. Moreover, since our surrogate is differentiable by design, it paves the way for methods that estimate both individual-level parameters and state variables.

To achieve this goal, our framework relies on a novel combination of graph neural networks and diffusion models. Diffusion models [14] were first introduced in the context of image generation,

where they demonstrated impressive generation capabilities [7], and were then applied to other domains [18]. A number of works addressed graph data [21], for example in molecule modeling [15] and protein structure generation [2]. However, these works focus on the generation of graphs, while our architecture learns to generate random samples that are *conditioned* on information found on a graph. To the best of our knowledge, our work is the first application of this generative framework to individual behavior modeling in simulation systems, such as ABMs.

## 3  Methods

Denoting the set of agents by $A$, let each agent $i \in A$ at discrete time $t$ be described by a state vector $\mathbf{Z}_t^{(i)}$, which may include both continuous and categorical features. Given the ABM parameters $\Theta$, the update rule of $\mathbf{Z}_t^{(i)}$ follows a stochastic transition process $P_\Theta$ given by

$$\mathbf{Z}_{t+1}^{(i)} \sim P_\Theta(\mathbf{Z}_{t+1}^{(i)}|\mathbf{Z}_t^{(i)}, \{\mathbf{Z}_t^{(j)}\}_{j \in N_t^{(i)}}), \tag{1}$$

where $N_t^{(i)}$ is the set of agents interacting with agent $i$ at time $t$, inducing a (time-varying) interaction graph $G_t = (A, E_t)$ that we assume to be known. This formulation focuses on *individual* agents, capturing not the dynamics of the entire system, but the evolution of each agent over time. In this way, it makes the two core ingredients of ABMs explicit: *(i) relational structure* via local neighborhoods $N_t^{(i)}$; *(ii) stochasticity* in the choice of next states.

To effectively model these components in the same individual-level view, we leverage respectively *(i) message-passing GNNs* [10], which model the relationship between the evolution of an agent's state and the state of its neighbors on the graph; *(ii) conditional diffusion models* [44], generative architectures well-suited to learning complex, multimodal distributions, allowing us to capture the intrinsic stochasticity of agent behavior.

Our proposed method, dubbed Graph Diffusion Network (GDN), combines these two components into a single architecture. Together, these components let us learn both how any agent state is affected by its neighbors on the graph, and the inherent randomness driving agent dynamics, yielding a surrogate that can both emulate the original ABM and be differentiated.

**Overview**. To learn the distribution $P_\Theta$, the training phase requires observations of different outcomes given the same starting conditions. To do so, in our framework, we use the original ABM to generate a data set as a *ramification* of possible states, namely $(\mathbf{Z}_t^{(i)}, \{\mathbf{Z}_t^{(j)}\}_{j \in N_t^{(i)}}) \longrightarrow \mathbf{Z}_{t+1}^{(i)}$ (see Figure 1). Our Graph Diffusion Network then approximates the stochastic kernel $P_\Theta$ by integrating a Message-Passing GNN with a Conditional Diffusion Model, of learnable parameters $\omega$ and $\phi$ respectively. The GNN aggregates each agent's state $\mathbf{Z}_t^{(i)}$ and its neighbors' states $\{\mathbf{Z}_t^{(j)}\}_{j \in N_t^{(i)}}$ via permutation-invariant message and readout functions to produce an interaction embedding $\mathbf{g}_t^{(i)}$. This embedding acts as a compact representation of the information coming from $i$'s neighbors at $t$, affecting the distribution of possible states of agent $i$ at time $t+1$. As such, it is passed to the diffusion model: conditioned on $\mathbf{Z}_t^{(i)}$ and $\mathbf{g}_t^{(i)}$, the diffusion model learns to transform a sample of Gaussian noise into a possible instance of the next state $\mathbf{Z}_{t+1}^{(i)}$. By minimizing the standard denoising loss over all observed transitions, this hybrid architecture captures both the graph-structured interactions and the inherent stochasticity of agent dynamics. The trained model $GDN_{\phi,\omega}$ is therefore able to generate, given a graph $G_t$ of interacting agents and the state of each one $\mathbf{Z}_t^{(i)}$, a sequence of possible next states $\mathbf{Z}_{t+1}^{(i)}$. The consecutive application of $GDN_{\phi,\omega}$ allows for reproducing the behavior of the original model. We now describe in detail each of these components.

**Message-passing GNN.** The GNN operates on the provided interaction graph $G_t = (A, E_t)$, that we assume to be known or to be computable from $\mathbf{Z}_t$ (e.g., in the Schelling model, the position of the agents determines who interacts with whom). For each agent $i$, the GNN aggregates its state $\mathbf{Z}_t^{(i)}$ together with each neighbor's state $\mathbf{Z}_t^{(j)}$ via a permutation-invariant operator $\bigoplus$, and then feeds the concatenated result through an MLP $f_\omega$ [10], that is $\mathbf{g}_t^{(i)} = f_\omega\Big(\mathbf{Z}_t^{(i)}, \bigoplus_{j \in N_t^{(i)}}(\mathbf{Z}_t^{(i)}, \mathbf{Z}_t^{(j)})\Big)$. The resulting vector $\mathbf{g}_t^{(i)}$ captures how $i$'s local neighborhood influences its next-state distribution. In

practice, for the operator $\bigoplus$, we used sum aggregation for Predator-Prey and mean aggregation for Schelling, since the latter's dynamics depend on the degree. Minimal experimental evaluation can guide practitioners toward the most suitable operator.

**Conditional diffusion model.** The diffusion model then learns the distribution over future states given this output from the graph and the state of a given agent. Diffusion models do so by reversing a fixed Gaussian noising process [14]. The obtained denoising process, indexed by $\tau \in \{\tau_{\max}, \ldots, 0\}$, starts from a sample of Gaussian noise $\mathbf{x}_{\tau_{max}} \sim \mathcal{N}(\mathbf{0}, \mathbf{I})$ and in a sequence of denoising diffusion steps transforms it into a possible next state $\mathbf{x}_0 \approx \mathbf{Z}_{t+1}^{(i)}$. In this setting, we denote the general latent $\mathbf{x}_\tau$ as $\tilde{\mathbf{Z}}_{t+1}^{(i)}(\tau)$. Each step of this process receives as input (i) the agent's current state $\mathbf{Z}_t^{(i)}$, (ii) its interaction embedding $\mathbf{g}_t^{(i)}$, and (iii) a sinusoidal positional embedding of $\tau$. These inputs are first transformed by MLPs to form the condition vector $\mathbf{c}_t^{(i)}$. Then, a feed-forward network $\phi$ is trained to predict the noise residual $\epsilon_\phi$, i.e., the change to apply to the input $\tilde{\mathbf{Z}}_{t+1}^{(i)}(\tau)$ to continue the denoising process.

**Ramification data set.** Given these two components, our framework uses the original ABM to produce a *ramifications* data set (see Figure 1). Such data set follows one main branch that specifies the evolution of the ABM, and multiple alternative stochastic evolutions of each time step from time $t$ to time $t + 1$. This method makes it possible to expose the model to multiple stochastic successors from identical conditioning context, while avoiding exponential growth in the number of histories. Starting from an initial configuration $\mathbf{Z}_0 = \left(\mathbf{Z}_0^{(1)}, \ldots, \mathbf{Z}_0^{(n)}\right)$, we recursively simulate $R + 1$ child configurations at each time step $t$, yielding $\{\mathbf{Z}_{t+1}[r]\}_{r=0,\ldots,R}$. We designate the branch $r = 0$ as the *main branch* $\{\mathbf{Z}_t[0]\}_{t=0,\ldots,T-1}$, from which we extract the conditioning tuples $\left(\mathbf{Z}_t^{(i)}, \{\mathbf{Z}_t^{(j)}\}_{j \in N_t^{(i)}}\right)$ for all agents $i$. The remaining $R$ sibling branches at each $t$ supply the target next states $\mathbf{Z}_{t+1}^{(i)}$, ensuring that each context yields multiple outcomes.

**Learning procedure.** Our framework uses these data sets to train the Graph Diffusion Network. It minimizes the expected denoising loss over the outcomes observed in the ramification data (see Algorithm 1). At each training iteration, it uniformly samples a time index $t$ and extracts the conditioning pair $(\mathbf{Z}_t^{(i)}, \{\mathbf{Z}_t^{(j)}\})$ from the main branch $\mathbf{Z}_t[0]$. We compute the interaction embedding $\mathbf{g}_t^{(i)}$ via Equation (3), then draw a diffusion step $\tau$ to form the condition vector $\mathbf{c}_t^{(i)}$, and uniformly select one of the $R$ next-state realizations to obtain the target $\mathbf{Z}_{t+1}^{(i)}$. Finally, we minimize the denoising loss in Equation (5) by backpropagating through both the diffusion model and the GNN. More details about the architecture,

---

**Algorithm 1: Training Procedure**

1: **repeat**
2:      $t \sim \mathrm{Uniform}(0, \ldots, T-1)$
3:      $\mathbf{Z}_t^{(i)}, \{\mathbf{Z}_t^{(j)}\}_{j \in N_t^{(i)}} \leftarrow \mathbf{Z}_t[0]$
4:      $\mathbf{g}_t^{(i)} = f_\omega(\mathbf{Z}_t^{(i)}, \bigoplus_{j \in N_t^{(i)}}(\mathbf{Z}_t^{(i)}, \mathbf{Z}_t^{(j)}))$
5:      $\tau \sim \mathrm{Uniform}(1, \ldots, \tau_{max})$
6:      $\tau_{emb} = \mathrm{SinusoidalPositionEmbedding}(\tau)$
7:      $\mathbf{c}_t^{(i)} = \mathrm{MLP}(\mathbf{Z}_t^{(i)}) + \mathrm{MLP}(\mathbf{g}_t^{(i)}) + \mathrm{MLP}(\tau_{emb})$
8:      $r \sim \mathrm{Uniform}(1, \ldots, R)$
9:      $\mathbf{Z}_{t+1}^{(i)} \leftarrow \mathbf{Z}_{t+1}[r]$
10:     $\epsilon \sim \mathcal{N}(\mathbf{0}, \mathbf{I})$
11:     **Optimizers step over all** $i \in A$
12:     $\nabla_{\phi,\omega}||\epsilon - \epsilon_\phi(\tilde{\mathbf{Z}}_{t+1}^{(i)}(\tau), \mathbf{c}_t^{(i)})||^2$
13: **until** convergence

---

and a discussion on computational costs and scalability can be found in Supplementary Section A.

## 4 Experiments

In this section, we present different experiments to assess and demonstrate our framework's ability to learn micro-level agent behaviors and faithfully reproduce emergent system-level dynamics. We evaluate our Graph Diffusion Network on two canonical agent-based models: the Schelling's segregation model and the Predator–Prey ecosystem, presented in Section 2 and detailed in Supplementary Section B. We test both its *micro-level* and *macro-level* fidelity. At the micro level, we measure how well the surrogate reproduces the conditional next-state distribution of each agent under identical context on an out-of-training ramification data set. At the macro level, we assess whether the surrogate, once trained on the first $T_{\mathrm{train}} = 10$ timesteps, can accurately reproduce the subsequent $T_{\mathrm{test}} = 25$ timesteps of aggregate summary statistics.

Because no existing method directly accepts graph-structured agent states and outputs per-agent state distributions, there are no directly applicable baselines for our approach. Existing surrogates typically operate at the macro level, learning mappings from parameters to aggregate outcomes rather than reproducing full system dynamics. However, such approaches, including standard time-series models such as AR(1), fail to capture non-monotonic or cyclic patterns (e.g., predator–prey oscillations) and do not generalize beyond in-sample dynamics, as we show in Supplementary Section C.5. Therefore, we evaluate against *two ablated variants*. The first replaces the GNN embedding with a flat concatenation of all agent states, removing relational structure. The second keeps the GNN but removes the diffusion component, predicting deterministic next states instead. Both ablations are trained on the same ramified datasets and under identical protocols.

In the remainder of this section, we first describe the experimental setup, including dataset generation, model variants, and evaluation metrics. We then present a qualitative analysis of emergent patterns, followed by a comprehensive quantitative comparison. All implementation and reproducibility details are provided in the Supplementary Materials. Full code to reproduce our experiments is available at http://github.com/fracozzi/ABM-Graph-Diffusion-Network.

## 4.1 Experimental design

**Ablation.** Our core hypothesis is that both relational structure and stochastic modeling are crucial for accurate ABM surrogates. We consider therefore two possible ablations. In the first, we remove the message-passing GNN and replace the interaction graph with a flat concatenation of all agents' state vectors—this isolates the impact of neglecting agent interactions. The second drops the diffusion component entirely, yielding a purely deterministic model similar in spirit to prior GNN-based approaches for deterministic automata [13]. In this ablated version, we predict the next agent state by minimizing the mean squared error (MSE) with respect to the true next agent state. Details about the architecture can be found in Supplementary Section A. These ablations allow us to measure the improvement achieved by combining relational and stochastic modeling.

**Agent-based models.** We evaluate our approach on the two ABMs described in Section 2 as case studies. The first is the Schelling segregation model, in which $n$ agents occupy cells on a two-dimensional grid. Each agent has a fixed binary "color" and a position on the grid. At each timestep, an agent is considered *happy* if the proportion of its (up to eight) immediate neighbors sharing its color exceeds a tolerance threshold $\xi$; otherwise, it is *unhappy* and relocates to a randomly selected empty cell; thus, the interaction graph $G_t$ links each agent to its neighbors at time $t$. We adopt the standard NetLogo implementation of this model [42]. The second is a predator–prey ecosystem model, where agents belong to one of two species (predator or prey), inhabit grid cells, and cycle through life phases—Unborn, Alive, Pregnant, and Dead. At each timestep, an Alive agent may move to a neighboring cell, reproduce (becoming Pregnant), or die, with probabilities specified by a parameter matrix $\Psi$ and conditioned on the local presence of predators or prey [39, 43]. Pregnant agents revert to Alive after giving birth; Unborn agents become Alive if their parent is Pregnant; and Dead agents remain inactive. Here, $G_t$ links Alive neighboring agents, with specific rules for Pregnant and Unborn agents. See Supplementary Section B for more details. In both ABMs, each agent's full state at time $t$ comprises its position, type (color or species), and, for the predator-prey ABM, its life phase. Together, these two models span both simple relocation dynamics and richer birth–death interactions, providing diverse testbeds for our surrogate.

**Micro evaluation metrics.** To quantify how faithfully our surrogate captures individual agent behavior, we compare its predicted conditional next-state distributions against the ABM's true stochastic transitions using the Earth Mover's Distance (EMD) [35]. We extend the ramification dataset beyond the training horizon and generate corresponding datasets for both the surrogate and the ablation models. The EMD is then computed as the mean value across timesteps and individual agents. In the Schelling ABM, the EMD compares the distribution of agent positions. This directly measures the surrogate's ability to relocate *unhappy* agents correctly, while keeping *happy* agents stationary. In the predator-prey model, we treat the agent's categorical life phase as the random variable and compute the EMD over its four-state distribution. This metric captures both deterministic transitions (e.g., Unborn → Alive, Dead → Dead) and stochastic, interaction-driven transitions (e.g., Alive → Dead, Alive → Pregnant).

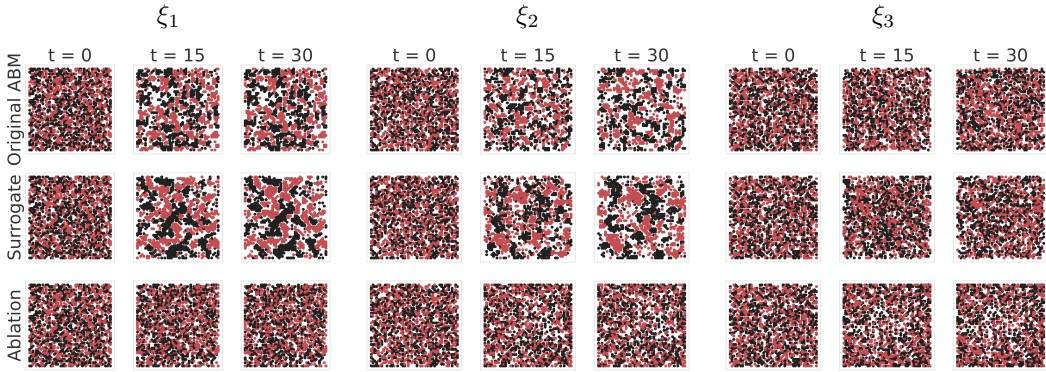

Figure 2: Evolution of the position of black and red agents in the Schelling model, for three simulation runs, one for each of the considered tolerance thresholds $\xi_1 = 0.625$, $\xi_2 = 0.75$, $\xi_3 = 0.875$ (left, center, and right panel). We compare the ground truth (top row) with our surrogate (middle row) and with the best-performing ablation (according to sMAPE, bottom row). For each panel and model, we show three time steps: $t = 0$ (initial conditions, same for each column but kept for clarity), $t = 15$, and $t = 30$.

**Macro evaluation metrics.** Next, we test whether agent-level predictions translate into faithful reproduction of emergent, system-level behavior. For each model, we track a summary statistic over time: the number of happy agents in Schelling, and the number of active (i.e. Alive and Pregnant) agents in the predator–prey ecosystem. Reusing the same ramification branches as in training would offer little new information, since different stochastic branches from the same state tend to produce very similar macroscopic trajectories. Instead, we generate a fresh ensemble of main-branch simulations (100 independent runs) beyond the training horizon. We then compute the symmetric mean absolute percentage error (sMAPE) between the mean ground-truth trajectory and the mean surrogate-predicted trajectory across this ensemble, providing a quantitative measure of the surrogate's ability to capture oscillations and steady-state behavior truly out-of-sample.

**Experimental set-up.** We consider three parameter combinations $\xi$ for the Schelling ABM, each producing distinct segregation outcomes, and four $\Psi$ combinations for the predator-prey ABM, reflecting different oscillatory patterns in the population dynamics. For each ABM and parameter setting, we simulate $T_{\text{train}} = 10$ main-branch steps with $R = 500$ stochastic branches per step, yielding the training ramification as in Figure 1. For macro-evaluation, we run 100 independent main-branch simulations to calculate sMAPE. For micro-evaluation, we generate an out-of-sample ramification dataset of $T = 25$ timesteps. We train both surrogate and ablations for 100 epochs using Adam with learning rate $10^{-5}$ for the diffusion model and Adam with learning rate $2 \cdot 10^{-5}$ for the GNN, batch size equal to number of agents, and diffusion hyper-parameters $\tau_{\text{max}} = 100$ (more information in Supplementary Section A).

### 4.2 Results

To build intuition, we first qualitatively compare the surrogate and its ablated variant on their ability to reproduce key emergent patterns of agent-based dynamics. We then consolidate these insights with a comprehensive quantitative evaluation using the macro- and micro-level metrics introduced in the previous section. We report a selection of results in this Section; more in Supplementary Section C.

**Reproducing emergent segregation.** Let us first consider the Schelling ABM, under the configurations $\xi_1 = 0.625$, $\xi_2 = 0.75$, $\xi_3 = 0.875$. Figure 2 illustrates how the ground-truth ABM (top row) progresses from a randomized initialization to structured, segregated communities for the first two configurations $\xi_1, \xi_2$, while it remains unsegregated for $\xi_3$. At the first two tolerance levels, in fact, the agents gradually self-organize into distinct clusters, with segregated communities clearly emerging around $t = 20$ (see Supplementary Section C). The middle row represents the evolution of the system according to our surrogate model: we initialize the system with the same starting condition, and then we iterate giving the current state $\mathbf{Z}_t$ to our model, and using one sample of

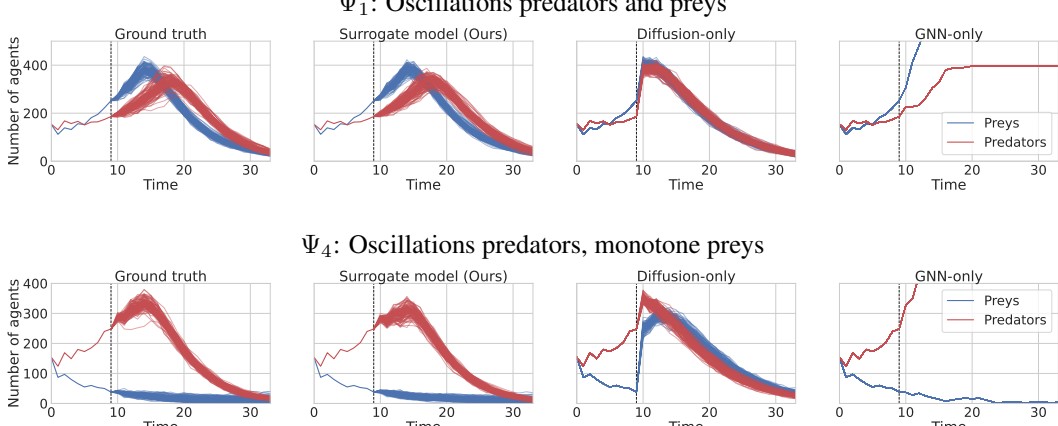

Figure 3: Forecasting macro-level summary statistics (here, the number of alive preys and predators over time), starting from the last condition seen in training, for 100 independent simulation runs, under configuration $\Psi_1$ (oscillations for both predators and preys, top) and $\Psi_4$ (oscillations only for predators, bottom). From left to right: original ABM simulations, surrogate, diffusion-only ablation, GNN-only ablation. The dashed vertical line indicates the end of the training phase for the surrogate and ablation models.

the generated output as the next state $\mathbf{Z}_{t+1}$. We observe that our surrogate exhibits a qualitatively similar pattern of cluster formation over time, distinct for each of the three configurations. Instead, the ablation models fail to meaningfully relocate agents. The best-performing one, according to sMAPE, the diffusion-only model, largely maintains a random configuration.

**Reproducing emergent oscillations in predator-prey dynamics.** Next, we consider the Predator-Prey ecological model. Figure 3 overlays 100 trajectories of prey and predator populations starting from the same state at the end of training, comparing the stochastic trajectories from the ground-truth model with those obtained by our surrogate and by the ablation. For both configurations, the surrogate and the ablation models are trained only with the initial time steps (up to the dashed line in the plots). Under the parameter set $\Psi_1$, the ground-truth ABM (top-left plot) exhibits classical Lotka–Volterra oscillations: a rise in prey growth drives a delayed increase in predators, which then triggers prey decline and a subsequent predator decline. Under $\Psi_4$, instead, only predators show a rise and decay, while preys only decline (bottom-left plot). We observe that the surrogate (second column) accurately captures both the phase lag and amplitude of these oscillations, while the diffusion-only ablation (third column) collapses to near-monotonic trends. The GNN-only ablation (fourth column), besides following a completely deterministic dynamic, completely diverges from the ground truth. The number of preys under $\Psi_1$, for instance, quickly reaches a plateau at a value of 750 (not shown), almost twice the real one. We perform the same analysis for alternative parameterizations $\Psi_2$, $\Psi_3$ (included in Supplementary Section C) that show different types of dynamics, as the population of predators and/or preys may exhibit monotonic extinctions. In all cases, the surrogate faithfully reproduces monotonic declines or single-peak dynamics and both ablations fail. We also observe (figures in Supplementary Section C) that the surrogate recreates the rich spatial patterns of predator–prey clusters, also seen in similar settings in evolutionary game theory [27].

**Quantitative results.** Now we present the results of a quantitative analysis, systematizing the previous comparisons. Here, each comparison with the ground truth is quantified using one of the metrics presented in the previous subsection, i.e. Earth Mover's Distance (EMD) for the micro-level comparisons, and sMAPE for the macro-level ones. Figure 4 summarizes the results of our experiments: the left panel shows the microscopic evaluation of both our surrogate model and the ablated variant, while the right panel presents the macroscopic evaluation results.

For the Schelling model, we observe that, on the micro level, the surrogate's mean EMD is lower than the diffusion-only ablation's mean EMD in all cases. The differences between the surrogate and the diffusion-only ablation are less pronounced at the thresholds $\xi_1$ and $\xi_3$. At $\xi_1$ (few *unhappy* agents) behavior is almost entirely deterministic and agents rarely move, while at $\xi_3$ (almost all

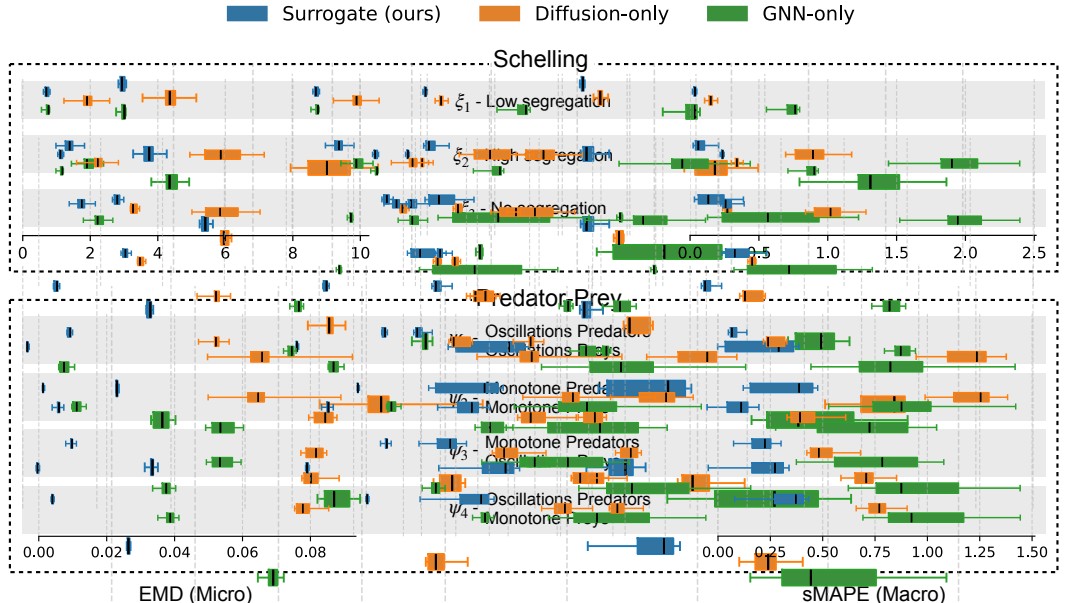

Figure 4: Errors obtained by the proposed approach (*Surrogate*) and by the naive baselines (*Diffusion-only* and *GNN-only* ablation models) in four different tasks. In the first column, error is measured as the EMD between the true and predicted distribution of individual (micro-level) behavior, i.e. predicting the next state of each agent from the previous one. In the second column, error is measured as the difference (sMAPE) in system-level quantities, i.e. comparing the true values of the number of agents with a given state with the one predicted by our model when trained on a fraction of the initial time steps (as in Figure 3). In the first row, we test three configurations of the Schelling model; in the second row, we compare four configurations of the Predator-Prey model.

agents *unhappy*) behavior is uniformly random, so even a flat, "always-move" or "never-move" rule yields near-optimal predictions in these two cases. In contrast, at the intermediate threshold $\xi_2$, where roughly half the agents are *unhappy*, the difference between the surrogate's and the diffusion-only ablation's EMD is more pronounced. A similar pattern is observed in the macroscopic evaluation. The surrogate's sMAPE remains below 0.2, whereas the diffusion-only ablation fails to distinguish *happy* from *unhappy* cases, resulting in large macro-level errors. In the GNN-only ablated model, at the micro-level EMD increases proportionally with the level of stochasticity introduced by the parameters, with EMD values similar to the surrogate for $\xi_1$ (the most deterministic) and much higher for $\xi_3$ (the most stochastic). At the macro-level, we observe large errors, with sMAPE always higher than the surrogate. These gaps confirm that only the full model, with explicit graph-based interaction modeling and stochasticity, can learn the conditional relocation rule critical in balanced regimes.

For the Predator–Prey model, regarding micro-level behavior, our surrogate achieves a low EMD from the ground truth on average, and it consistently outperforms both ablation models. These results confirm that our model is able to faithfully reproduce the complex dynamics of this ABM even at the individual agent-level (thus explaining Figure 3). The most successful case is $\Psi_2$, where our surrogate exhibits a near-zero difference from the ground truth. In fact, in this configuration, most agents follow deterministic update rules (e.g., *dead $\rightarrow$ dead*), which are perfectly recovered by our model, but not by the diffusion-only ablation — which also obtains worse results on stochastic rules, as shown in Supplementary Section C.4. Instead, the GNN-only ablation performs well only in those deterministic cases, but fails in all the others. At the macro level as well, the surrogate consistently outperforms the ablation, generally achieving low error. The best result is obtained with $\Psi_1$, the most complex dynamics, where the surrogate achieves an average sMAPE of approximately 0.08. This configuration produces two distinct population peaks, and the surrogate faithfully reproduces both their timing and amplitude (Figure 3). The worst result is obtained with $\Psi_2$, as this configuration is almost monotonic and dominated by long, near-zero tails that are noisy at very small scales, making them difficult for any model to reproduce.

# 5  Discussion

We introduced Graph Diffusion Networks, a differentiable surrogate for agent-based models that combines graph neural networks to model agent-level interactions with diffusion models to capture stochasticity. Our experiments on the Schelling segregation model and a Predator–Prey ecosystem show that this approach not only accurately reproduces individual-level transition distributions, but also faithfully captures emergent, system-level dynamics beyond the training horizon.

**Limitations.** Our approach is limited by our assumptions about the characteristics of the ABM to emulate. First, the interaction graph is assumed to be fully known. Future work might remove this limitation by estimating such a graph directly from available data. However, the estimation of a latent interaction graph is a follow-up challenge, for which our GNN-based approach represents a necessary first step. Second, highly sophisticated ABMs may include features not addressed in our framework - such as all-to-all interactions, multiple rounds of decision-making, or sequential stochastic events within a single time step. Capturing these dynamics may require extending our architecture to incorporate sequential or hierarchical components. While our method may not yet fully generalize to such settings, our findings demonstrate that building surrogates capable of replicating individual-level behavior is both feasible and effective, laying the groundwork for broader applications.

**Future work.** Building on this foundation, the differentiability of our surrogate opens up a range of powerful applications. It enables the use of gradient-based methods for any optimization task, such as policy optimization [1]. It allows for efficient calibration of macro parameters by treating key parameters as additional inputs to the neural network. Most importantly, our approach naturally allows for the estimation of micro (i.e., agent) level variables — a challenge for ABMs, that often requires the ad hoc development of handcrafted probabilistic models [24, 25]. In fact, our model already contains such parameters expressed as agents' individual states ($\mathbf{Z}_t^{(i)}$), something typically not available in ABM surrogates [11]. Moreover, our method can in principle be applied directly to real-world datasets whenever sufficient observations of comparable agent–context pairs and transitions are available. In doing so, our framework helps make ABMs more data-driven and empirically grounded, with promising applications in several scientific domains, such as economics [29], epidemiology [12], sustainability [19], urban science [4], and ecology [38].

# Acknowledgments

The authors wish to thank Daniele Grattarola and Federico Cinus for insightful early discussions that supported the initial development of this work. We also thank Alberto Novati for his contribution to the early draft of the code for the original ABM of the predator-prey system.

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

# Learning Individual Behavior in Agent-Based Models with Graph Diffusion Networks Supplementary Material

## A  Neural models and training details

In this section, we provide a detailed overview of the core components of the Graph Diffusion Network (GDN) and its methodology. We begin by introducing the diffusion process (A.1), which defines the diffusion process to be reversed to generate future agent states starting from a sample of Gaussian noise. Next, we detail the Graph Diffusion Network architecture (A.2) and its components. We then discuss the loss and optimization strategy (A.3), covering the training objectives and gradient flow between the diffusion model and graph components. Following this, we outline the generation algorithm (A.4), where the iterative denoising process generates future agent states. Finally, we provide a discussion on the computational costs and scalability of our methodology (A.5).

### A.1  Diffusion process

Our diffusion model is designed to generate future agent states $\mathbf{Z}_{t+1}^{(i)}$ by reversing a known Gaussian noising process (i.e. the *forward process*) through a set of latents $\tilde{\mathbf{Z}}_{t+1}^{(i)}(\tau)$ indexed by $\tau \in \{\tau_{max}, ..., 0\}$. The forward process is a fixed Markov chain that gradually adds Gaussian noise to the input $\mathbf{Z}_{t+1}^{(i)}$ according to a previously defined variance schedule $\beta_\tau$. Each latent is given by:

$$\tilde{\mathbf{Z}}_{t+1}^{(i)}(\tau) = \sqrt{\bar{\alpha}_\tau}\mathbf{Z}_{t+1}^{(i)} + \sqrt{1 - \bar{\alpha}_\tau}\epsilon, \quad \epsilon \sim \mathcal{N}(\mathbf{0}, \mathbf{I}) \tag{2}$$

where $\alpha_\tau := 1 - \beta_\tau$ and $\bar{\alpha}_\tau := \prod_{s=1}^{\tau} \alpha_s$. For our model, we selected a *cosine* variance schedule:

$$\beta_\tau = \beta_{start} + \frac{1}{2}(\beta_{end} - \beta_{start})(1 - cos(\frac{\tau}{\tau_{max}}\pi)) \tag{3}$$

with $\beta_{start} = 10^{-4}$ and $\beta_{end} = 0.02$. This choice ensures that $\beta_\tau$ increases more gradually at the beginning of the forward process, retaining more of the original input information, and at the end of the forward process. We note that, in preliminary trials, it showed to be more stable in our scope with small input dimensions compared to the *cosine* variance schedule proposed by [48] in the context of image generation.

### A.2  Graph Diffusion Network architecture

The primary input of the conditional diffusion model inside the Graph Diffusion Network is the latent $\tilde{\mathbf{Z}}_{t+1}^{(i)}(\tau)$, a noised version of $\mathbf{Z}_{t+1}^{(i)}$ given by equation (2). In general, not all variables contained in $\mathbf{Z}_{t+1}^{(i)}$ are time-dependent, and some remain stationary through time (e.g. color in Schelling and kind in Predator-Prey, see Supplementary subsections B.1, B.2). We only include the time-dependent features (or dynamical features) in the input of the diffusion model, as they are the ones that evolve over time and need to be predicted. The output of the diffusion model is the denoising step $\epsilon_\phi(\tilde{\mathbf{Z}}_{t+1}^{(i)}(\tau), \mathbf{c}_t^{(i)})$ introduced in Section 3, which has the same size as the input. Thus, the neural network follows a symmetrical structure with hidden layers of increasing width in the first half, and decreasing width in the second. The condition vector $\mathbf{c}_t^{(i)}$ is applied to the hidden layers of the neural network by applying an activation function, performing a linear operation to match the width of the layer and summing element-wise with the hidden layer. To increase stability, hidden layers are first normalized and there is a residual connection after conditioning has been applied. All details of the architecture of the conditional diffusion model are reported in Table 1.

The Message Passing GNN takes in input the entire agent state $\mathbf{Z}_t^{(i)}$ as node features. The messages correspond to the node features and are aggregated by an aggregation function such as *sum* or *mean* value. The choice of the aggregation function depends on the ABM to be reproduced. In general, *sum* is a suitable choice, as the MLP $f_\omega$ will capture the behavior rules of the agents. However, for ABMs where the behavior of agents is influenced by the node degree, as in the case of Schelling, *mean* is a more appropriate choice. All details of the architecture of the Message Passing GNN are reported in Table 1.

To make the network more stable, all features are scaled. In particular, agent states $\mathbf{Z}_t^{(i)}$ can contain both numerical and categorical features. Numerical features are scaled to the interval $[-1, 1]$. In our experiments, we scaled numerical features with a standard scaler. After generation, they are scaled back to their original domain and, for integer numerical features, a binning function is applied afterwards. Categorical features are one-hot encoded.

Table 1: Architecture and training details of the Graph Diffusion Network

| Component | Details |
|---|---|
| **Conditional diffusion model** | |
| Input dimension | `dynamical_features_dim` |
| Hidden layers | [128, 256, 1024, 1024, 256, 128] |
| Output dimension | `dynamical_features_dim` |
| Activation function | LeakyReLU (slope = 0.1) |
| MLP time embedding | Linear(256) $\rightarrow$ Act $\rightarrow$ Linear(256) |
| MLP current state | Linear(256) $\rightarrow$ Act $\rightarrow$ Linear(256) $\rightarrow$ Act $\rightarrow$ Linear(256) |
| MLP graph embedding | Linear(256) $\rightarrow$ Act $\rightarrow$ Linear(256) $\rightarrow$ Act $\rightarrow$ Linear(256) |
| Condition block in hidden layers | LayerNorm $\rightarrow$ Linear(`dim_out`) $\rightarrow$ Sum(Lin(Act(condition))) $\rightarrow$ Linear(`dim_out`) $\rightarrow$ Residual connection |
| Weights initialization | Xavier uniform |
| Optimizer | Adam |
| Learning rate | $10^{-5}$ |
| **Message Passing GNN** | |
| Input dimension | $2 \times$ `agent_state_dim` |
| Hidden layers | [32, 64, 128] |
| Output dimension | 256 |
| Aggregation function | `sum` or `mean` |
| Activation function | LeakyReLU (slope = 0.1) |
| Message passing | Message$(x_j) = x_j$ |
| Weights initialization | Kaiming uniform |
| Optimizer | Adam |
| Learning rate | $2 \times$ `learning_rate_diffusion` |
| **Other details** | |
| $\tau_{max}$ | 100 |
| Batch size | Number of agents in the system |
| Number of epochs | 100 |

**GNN-only ablated model**   The only architectural change of the GNN-only ablated model is in the MLP following the aggregation pass of the GNN: its output dimension is set to the agent-state dimension rather than the embedding dimension. While the original GNN uses hidden layers of size [32, 64, 128], the ablated model adopts a symmetrical structure: [32, 64, 128, 128, 64, 32]. We trained for 100 epochs with the Adam optimizer (learning rate $2 \times 10^{-5}$) and a mini-batch size of 16.

## A.3 Loss and optimization

The learning objectives of the Graph Diffusion Network are the noise residuals $\epsilon_\phi(\tilde{\mathbf{Z}}_{t+1}^{(i)}(\tau), \mathbf{c}_t^{(i)})$ of the denoising diffusion process in $\tau = \tau_{max}, ..., 0$ used to generate $\mathbf{Z}_{t+1}^{(i)}$, given $(\mathbf{Z}_t^{(i)}, \{\mathbf{Z}_t^{(j)}\}_{j \in N_t^{(i)}})$.

The generative process is conditioned by the condition vector $\mathbf{c}_t^{(i)}$, which is learned by the network $\phi$ and is given by:

$$\mathbf{c}_t^{(i)} = MLP(\mathbf{Z}_t^{(i)}) + MLP(\mathbf{g}_t^{(i)}) + MLP(\tau_{emb}) \tag{4}$$

where $\tau_{emb}$ is the sinusoidal positional embedding of $\tau$ and $\mathbf{g}_t^{(i)}$ is the embedding produced by the Message Passing GNN of parameters $\omega$. More details on the three MLPs that form $\mathbf{c}_t^{(i)}$ are given in Table 1. The loss function for the denoising diffusion steps $\epsilon_\phi(\tilde{\mathbf{Z}}_{t+1}^{(i)}(\tau), \mathbf{c}_t^{(i)})$ is calculated as the expected value over all agents in the system $i \in A$, all ABM timesteps $t$, all $\tau$ and $\epsilon \sim \mathcal{N}(\mathbf{0}, \mathbf{I})$:

$$L(\phi, \omega) = \mathbb{E}_{i,t,\tau,\epsilon} \left[ ||\epsilon - \epsilon_\phi(\tilde{\boldsymbol{Z}}_{t+1}^{(i)}(\tau), \boldsymbol{c}_t^{(i)})||^2 \right] \tag{5}$$

The parameters to optimize are the parameters of the diffusion model $\phi$ and the parameters of the GNN $\omega$. At each training step, the loss in equation (5) is calculated over the batch made of all agents $i \in A$ and gradients are backward propagated. First, the optimizer for $\phi$ is applied, and then the optimizer for $\omega$. Thus, the GNN is trained by inheriting the gradients from the loss of the conditional diffusion model, through the learned condition representation $\mathbf{c}_t^{(i)}$.

## A.4 Generation

The generation of $\mathbf{Z}_{t+1}^{(i)}$ starts from the last latent of the denoising diffusion process, which is a sample of Gaussian noise $\tilde{\mathbf{Z}}_{t+1}^{(i)}(\tau_{max}) \sim \mathcal{N}(\mathbf{0}, \mathbf{I})$. The Message Passing GNN takes in input the current state $\mathbf{Z}_t^{(i)}$ and the states of its neighbors $\{\mathbf{Z}_t^{(j)}\}_{j \in N_t^{(i)}}$ and forms the embedding $\mathbf{g}_t^{(i)}$. Then, iteratively over $\tau = \tau_{max}, ..., 1$, the conditional diffusion model takes in input the sinusoidal positional embedding $\tau_{emb}$, the current agent state $\mathbf{Z}_t^{(i)}$ and the embedding vector $\mathbf{g}_t^{(i)}$, and forms the condition vector $\mathbf{c}_t^{(i)}$. Lastly, the previous latent $\tilde{\boldsymbol{Z}}_{t+1}^{(i)}(\tau - 1)$ is calculated given $\tilde{\boldsymbol{Z}}_{t+1}^{(i)}(\tau)$ and the output of the Graph Diffusion Network $\epsilon_\phi(\tilde{\boldsymbol{Z}}_{t+1}^{(i)}(\tau), \boldsymbol{c}_t^{(i)})$ as in lines 7-8 in Algorithm 1.

---

**Algorithm 1** Generation

---

1: $\mathbf{Z}_t^{(i)}, \{\mathbf{Z}_t^{(j)}\}_{j \in N_t^{(i)}} \leftarrow Data$
2: $\mathbf{g_t}^{(i)} = f_\omega(\mathbf{Z}_t^{(i)}, \bigoplus_{j \in N_t^{(i)}}(\mathbf{Z}_t^{(i)}, \mathbf{Z}_t^{(j)}))$
3: $\tilde{\mathbf{Z}}_{t+1}^{(i)}(\tau_{max}) \sim \mathcal{N}(\mathbf{0}, \mathbf{I})$
4: **for** $\tau = \tau_{max}, ..., 1$ **do**
5:     $\tau_{emb} = \text{SinusoidalPositionEmbedding}(\tau)$
6:     $\mathbf{c}_t^{(i)} = \text{MLP}(\mathbf{Z}_t^{(i)}) + \text{MLP}(\mathbf{g}_t^{(i)}) + \text{MLP}(\tau_{emb})$
7:     $\mathbf{z} \sim \mathcal{N}(\mathbf{0}, \mathbf{I})$, if $t > 1$ else $\mathbf{z} = 0$
8:     $\tilde{\mathbf{Z}}_{t+1}^{(i)}(\tau - 1) = \frac{1}{\sqrt{\alpha_\tau}}(\tilde{\mathbf{Z}}_{t+1}^{(i)}(\tau) - \frac{1-\alpha_\tau}{\sqrt{1-\bar{\alpha}_\tau}}\epsilon_\phi(\tilde{\mathbf{Z}}_{t+1}^{(i)}(\tau), \mathbf{c}_t^{(i)})) + \sigma_\tau \mathbf{z}$
9: **end for**
10: **return** $\tilde{\mathbf{Z}}_{t+1}^{(i)}(0) \approx \mathbf{Z}_{t+1}^{(i)}$

---

We set $\sigma_\tau = \sqrt{\frac{1-\bar{\alpha}_{\tau-1}}{1-\bar{\alpha}_\tau}\beta_\tau}$. This choice of $\sigma_\tau$ is optimal for deterministically set points [14], which is the case for some update rules in ABMs (e.g. happy agents in Schelling and dead agents in Predator-Prey). Alternatively, one can also choose $\sigma_\tau = \sqrt{\beta_\tau}$, which is more optimal for normally distributed points.

## A.5 Computational costs and scalability.

In the following subsection, we provide additional details and discussion on the computational costs and scalability of our proposed methodology. In particular, details on the hardware employed to run

our experiments, performance details with respect to the number of ramifications provided during training, as well as a discussion over the scalability of Graph Diffusion Networks to larger systems of agents, and on the interplay between the number of agents and the number of ramifications.

**Runtime characteristics.**   All experiments were run in a cloud-based server with 15 vCores, 180 GB of RAM, and an NVIDIA A100 80GB PCIe GPU. Execution times depend on the size of the datasets and GPU occupancy by other processes. In our experiments, where the training datasets were made of $R = 500$ ramifications over $T = 10$ timesteps with 2048 agents in the system for Predator-Prey and 1950 agents for Schelling, the training time over 100 epochs typically lasted around 1 hour (around 37 seconds per epoch). Generating a simulation of 25 timesteps for the entire system of agents takes roughly 7.5 seconds for both Predator-Prey and Schelling, around 0.3 seconds per timestep.

**Scalability with respect to number of ramifications.**   During training, each training step processes the entire system of agents for one timestep and one ramification. Therefore, the runtime grows linearly with respect to the number of ramifications or timesteps. We present results from 10 experiment runs trained on one of the Predator-Prey datasets of parameter $\Psi_1$, to compare training time, micro-level metric (EMD), and macro-level metric (sMAPE) as the number of ramifications provided during training increases in Figure 5 and Table 2. As expected, training time increases linearly with the number of ramifications included in the dataset used to train the model. Furthermore, micro-level metrics decrease as the number of ramifications increases, showcasing the benefit of providing a higher number of stochastic outcomes to the model during training to better fit the stochastic rules that govern the original ABM. Macro-level metrics are subject to higher variance and do not show a trend as clear as with the micro-level metrics, still the best results are reached with the highest number of ramifications.

**Scalability with respect to number of agents.**   The experiments covered in Section 4.2 are performed on mid-sized ABMs, with the total agent count ranging in the thousands (for Schelling there are 1950 agents in total and 2048 for Predator-Prey). Increasing the number of agents in the system (for example by increasing the grid size and keeping the density of agents constant) increases the training time sub-linearly. In fact, with mid-sized systems, we can train the network with batches that correspond to the whole set of agents. Thus, the number of training iterations per epoch does not change, but rather the size of the batch in input to the network. It should be noted that increasing the number of agents also increases the inference time (time required by the trained surrogate to produce a simulation for the whole system). We present results from 5 experiment runs trained on one of the Predator-Prey datasets of parameter $\Psi_1$, to compare training time, inference time, micro-level metric (EMD) and macro-level metric (sMAPE) as the total number of agents in the system increases by increasing grid size and keeping agent density constant in Figure 6 and Table 3. From our results, it can be noted that micro-level metrics slightly decrease as agent count increases, as well as macro-level metrics reach lower values in some of the experiments with higher agent count. We argue that increasing the number of agents in the system naturally increases the number of agent transitions available in the dataset, expanding the statistical coverage provided to the model to learn, and yielding slightly better results.

**Scalability interplay between number of agents and ramifications.**   Generating hundreds of futures per state can be costly for many-agent or long-horizon systems, but larger systems naturally provide more independent samples of similar conditions; thus, fewer ramifications can suffice for comparable statistical coverage. To evaluate the trade-off between data generation cost and predictive quality, we performed 5 experiment runs and trained our model on one Predator–Prey dataset (parameter set $\Psi_1$) with increasing agent counts and decreasing ramifications, keeping the training time per epoch approximately constant, and the total number of agent transitions roughly similar. All other hyperparameters match the main experiments. We observe from Table 4 that the errors (both sMAPE and EMD) remain close to the original results even when the number of agents grows substantially, keeping training time low thanks to a lower number of ramifications needed.

Table 2: Effect of ramifications $R$ on training time and performance.

| Ramifications | Time/epoch | sMAPE | EMD |
|---|---|---|---|
| 50 | $3.71 \pm 0.02$ s | $0.076 \pm 0.019$ | $0.0161 \pm 0.0003$ |
| 100 | $7.44 \pm 0.02$ s | $0.071 \pm 0.015$ | $0.0156 \pm 0.0004$ |
| 250 | $18.61 \pm 0.06$ s | $0.083 \pm 0.039$ | $0.0115 \pm 0.011$ |
| 500 | $37.17 \pm 0.12$ s | $0.062 \pm 0.030$ | $0.0088 \pm 0.0007$ |

Table 3: Scaling with grid size, number of agents with fixed number of ramifications ($R = 500$).
*: Mean time to produce a simulation of the whole system for $t = 25$ timesteps.

| Grid size | Agents | Time/epoch | Inference time* | sMAPE | EMD |
|---|---|---|---|---|---|
| 64 | 8192 | $108.58 \pm 0.30$ s | $35.32 \pm 2.03$ s | $0.051 \pm 0.026$ | $0.0082 \pm 0.0003$ |
| 48 | 4608 | $63.78 \pm 0.32$ s | $16.45 \pm 0.72$ s | $0.053 \pm 0.026$ | $0.0083 \pm 0.0005$ |
| 32 | 2048 | $37.20 \pm 0.12$ s | $7.44 \pm 0.01$ s | $0.065 \pm 0.033$ | $0.0092 \pm 0.0007$ |

Table 4: Scaling with grid size, number of agents, and ramifications while keeping training time and transitions roughly constant.

| Grid size | Agents | Ramifications | Time/epoch | sMAPE | EMD |
|---|---|---|---|---|---|
| 64 | 8192 | 180 | $38.88 \pm 0.18$ s | $0.066 \pm 0.027$ | $0.0095 \pm 0.0005$ |
| 48 | 4608 | 300 | $38.41 \pm 0.25$ s | $0.073 \pm 0.024$ | $0.0092 \pm 0.0005$ |
| 32 | 2048 | 500 | $37.20 \pm 0.12$ s | $0.065 \pm 0.033$ | $0.0092 \pm 0.0007$ |

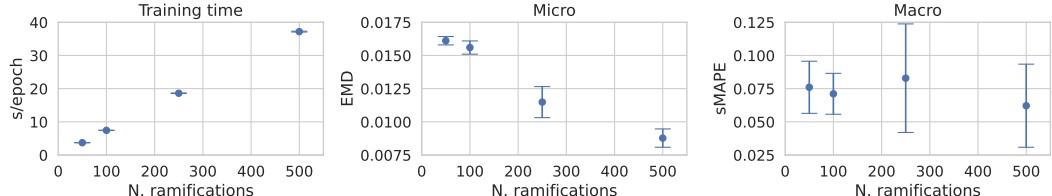

Figure 5: Training time, micro and macro metrics with respect to the number of ramifications provided during training for 10 experiments on one dataset of Predator-Prey with parameter set $\Psi_1$. Points indicate the mean value and error bars standard deviation over the 10 experiments.

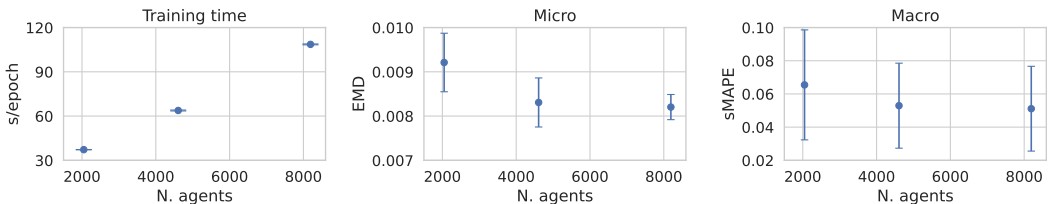

Figure 6: Training time, micro and macro metrics with respect to the total number of agents in the system for 5 experiments on one dataset of Predator-Prey with parameter set $\Psi_1$. Points indicate the mean value and error bars standard deviation over the 5 experiments.

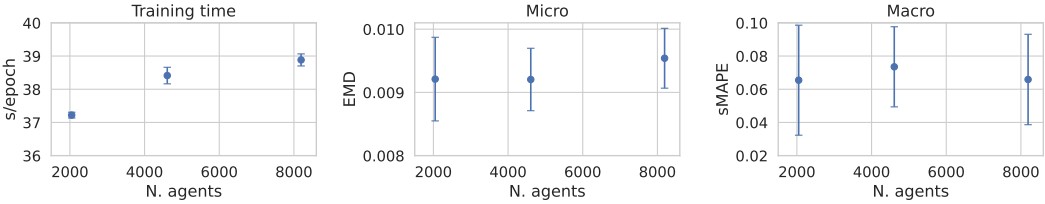

Figure 7: Training time, micro and macro metrics with increasing the total number of agents in the system and decreasing the number of ramifications to yield similar training times. Results are shown for 5 experiments on one dataset of Predator-Prey with parameter set $\Psi_1$. Points indicate the mean value and error bars standard deviation over the 5 experiments.

# B ABM Case Studies

## B.1 Schelling model

The **Schelling model of segregation** is a classic example to showcase the *emergence* property of ABMs. Agents $i \in A$ are placed in a 2-dimensional grid $L \times L$. Their state is given by their color (a binary variable such as *black* and *white*) and their position on the grid.

$$\mathbf{Z}_t^{(i)} = (C^{(i)}, x_t^{(i)}, y_t^{(i)})$$

$$C^{(i)} \in \{C_1, C_2\}, \quad x_t^{(i)}, y_t^{(i)} \in [0, L-1]$$

Each agent $i$ has a fixed color $C^{(i)}$, which remains constant over time, while their position on the grid may change. The set of agents that interact with agent $i$, denoted $j \in \mathcal{N}(i)$, includes those in the eight adjacent cells (Moore neighborhood) of $(x_t^{(i)}, y_t^{(i)})$.

The ABM mostly depends on a parameter $\xi \in [0, 1]$, representing the *intolerance* of the agents. If the fraction of neighbors $j \in \mathcal{N}(i)$ that share the same color as agent $i$ is *greater than or equal to $\xi$*, agent $i$ is considered *happy* and remains in its current position:

$$(x_{t+1}^{(i)}, y_{t+1}^{(i)}) = (x_t^{(i)}, y_t^{(i)}).$$

Conversely, if the fraction is *less than $\xi$*, agent $i$ is considered *unhappy* and moves to a randomly chosen empty cell on the grid. Thus, the update rule is deterministic when agents are happy, and stochastic when they are unhappy.

Algorithm 2 presents the pseudo-code of the ABM. Of particular interest are lines 15–25, which describe how agents relocate by searching for an empty cell on the grid. It is clear that this search process is not a simple draw from a probability distribution, as in the framework by Arya et al. [3], but a much more complex trial and error process. The pseudocode makes it clear that agents are more likely to relocate to nearby positions rather than distant ones, due to the way direction and distance are sampled. This spatial bias will be quantitatively confirmed in Figure 16.

## B.2 Predator-Prey model

The **Predator-Prey** ABM is a simulation model that captures the dynamics of interacting populations over time. We use a slightly adapted version of the model introduced in Ref. [39] (Algorithm 3 presents the pseudo-code of our ABM). Agents $i \in A$ occupy a two-dimensional grid of size $L \times L$. Each agent's state at time $t$ is given by its kind (either *Prey* or *Predator*), its life phase (*Unborn*, *Alive*, *Pregnant*, or *Dead*), and its position on the grid:

$$\mathbf{Z}_t^{(i)} = (K^{(i)}, f_t^{(i)}, x_t^{(i)}, y_t^{(i)})$$

$$K^{(i)} \in \{\text{Prey}, \text{Predator}\}, \quad f_t^{(i)} \in \{\text{Unborn}, \text{Alive}, \text{Pregnant}, \text{Dead}\}, \quad x_t^{(i)}, y_t^{(i)} \in [0, L-1]$$

The agent's kind $K^{(i)}$ is fixed over time, while the phase and position can evolve. The set of interacting agents $j \in \mathcal{N}^{(i)}$ consists of those located in the four cardinally adjacent cells to $(x_t^{(i)}, y_t^{(i)})$ (Von Neumann neighborhood).

In addition, each *Unborn* agent $i$ is assigned a parent agent $j$ of the same kind, provided $j$ is either *Alive* or *Pregnant*. This parent-child relationship governs the birth mechanism.

The dynamics are more complex than in Schelling's model of segregation. An *Alive* agent can transition to one of three states: remain *Alive*, become *Pregnant*, or become *Dead*. These transitions are stochastically determined. If the agent remains *Alive*, it moves at random to a cardinally adjacent cell. If it becomes *Pregnant*, it remains in place. If it becomes *Dead*, it loses its position on the grid. A *Dead* agent remains dead and off-grid. A *Pregnant* agent returns to being *Alive* in the same cell. An *Unborn* agent becomes *Alive* only if its assigned parent is currently *Pregnant*; otherwise, it remains *Unborn*.

This gives rise to a set of deterministic update rules:

$$\text{Dead} \to \text{Dead}, \quad \text{Pregnant} \to \text{Alive}, \quad \text{Unborn} \to \begin{cases} \text{Alive} & \text{if parent is Pregnant} \\ \text{Unborn} & \text{otherwise} \end{cases}$$

**Algorithm 2** Schelling Model of Segregation

---

**Require:** Agent set $A$, grid size $L$, tolerance $\xi$, max steps $T$, max distance $d_{\max}$, max trials $K$

1: **Initialize:** For each $i \in A$, sample $C^{(i)} \sim \text{Uniform}\{C_1, C_2\}$, $(x_0^{(i)}, y_0^{(i)}) \sim \text{UniformGrid}(L)$, and set $Z_0^{(i)} = (C^{(i)}, x_0^{(i)}, y_0^{(i)})$.

2: **for** $t = 0, \ldots, T-1$ **do**

3:      unhappy $\leftarrow \emptyset$                 ▷ Reset unhappy list

4:      **for all** $i \in A$ **do**

5:          $\mathcal{N}_t^{(i)} \leftarrow \{ j \in A : (x_t^{(j)}, y_t^{(j)}) \in \text{Moore}(x_t^{(i)}, y_t^{(i)}) \}$       ▷ Moore neighborhood

6:          $s \leftarrow |\{ j \in \mathcal{N}_t^{(i)} : C^{(j)} = C^{(i)} \}|, \quad n \leftarrow |\mathcal{N}_t^{(i)}|$      ▷ Same-color neighbors, all neighbors

7:          $r \leftarrow \begin{cases} s/n & n > 0 \\ 0 & n = 0 \end{cases}$            ▷ Similarity ratio

8:          **if** $r < \xi$ **then**             ▷ Agent $i$ is unhappy

9:              unhappy $\leftarrow$ unhappy $\cup \{i\}$

10:          **end if**

11:      **end for**

12:      **if** unhappy $= \emptyset$ **then**

13:          **break**                  ▷ Convergence

14:      **end if**

15:      **for all** $i \in$ unhappy **do**

16:          $(x_{t+1}^{(i)}, y_{t+1}^{(i)}) \leftarrow (x_t^{(i)}, y_t^{(i)})$

17:          **for** $k = 1, \ldots, K$ **do**

18:              $\theta \sim \text{Uniform}(0, 2\pi), \ d \sim \text{Uniform}(0, d_{\max})$    ▷ Random direction, distance up to $d_{\max}$

19:              $\Delta x \leftarrow \lfloor d \cos\theta \rfloor, \ \Delta y \leftarrow \lfloor d \sin\theta \rfloor$       ▷ Convert to grid movement

20:              $x^* \leftarrow (x_t^{(i)} + \Delta x) \bmod (L), \ y^* \leftarrow (y_t^{(i)} + \Delta y) \bmod (L)$     ▷ Wrap around border

21:              **if** $\neg \exists j \neq i : (x_t^{(j)}, y_t^{(j)}) = (x^*, y^*)$ **then**       ▷ Check if cell is empty

22:                  $(x_{t+1}^{(i)}, y_{t+1}^{(i)}) \leftarrow (x^*, y^*)$         ▷ Move to new location

23:                  **break**       ▷ Stop searching once valid position is found

24:              **end if**

25:          **end for**

26:      **end for**

27:      **for all** $i \in A \setminus$ unhappy **do**

28:          $Z_{t+1}^{(i)} \leftarrow Z_t^{(i)}$               ▷ Happy agents stay put

29:      **end for**

30: **end for**

---

And a set of stochastic update rules:

$$\text{Alive} \rightarrow \begin{cases} \text{Alive} & \text{(move)} \\ \text{Pregnant} & \text{(reproduce)} \\ \text{Dead} & \text{(die)} \end{cases}$$

All transitions are governed by a matrix $\Psi$, which specifies deterministic rules as probabilities equal to 1, and defines stochastic transitions through probabilities that depend on spatial inter-species interactions. We report the values of the matrix $\Psi$ that define our four experimental settings in Tables 5-8. For instance, a *Prey* that interacts with a *Predator* is more likely to die compared to a *Prey* that does not interact with a *Predator*. Similarly, a *Predator* that does not interact with a *Prey* is more likely to die compared to a *Predator* that does interact with a *Prey*, as it is more likely to starve. These spatial interactions are defined by proximity: an agent interacts with others located in its Von Neumann neighborhood (i.e., the 4-neighborhood).

Table 5: Transition matrix $\Psi_1$

|  | Die | Move | Turn pregnant | Turn alive | Stay dead | Stay unborn |
|---|---|---|---|---|---|---|
| Alive Pred + Prey | 0.15 | 0.45 | 0.40 | 0.00 | 0.00 | 0.00 |
| Alive Pred + No prey | 0.25 | 0.55 | 0.20 | 0.00 | 0.00 | 0.00 |
| Alive Prey + Pred | 0.30 | 0.45 | 0.25 | 0.00 | 0.00 | 0.00 |
| Alive Prey + No pred | 0.15 | 0.40 | 0.45 | 0.00 | 0.00 | 0.00 |
| Pregnant + Unborn child | 0.00 | 0.00 | 0.00 | 1.00 | 0.00 | 0.00 |
| Dead | 0.00 | 0.00 | 0.00 | 0.00 | 1.00 | 0.00 |
| Unborn + Npp* | 0.00 | 0.00 | 0.00 | 0.00 | 0.00 | 1.00 |

Table 6: Transition matrix $\Psi_2$

|  | Die | Move | Turn pregnant | Turn alive | Stay dead | Stay unborn |
|---|---|---|---|---|---|---|
| Alive Pred + Prey | 0.35 | 0.45 | 0.20 | 0.00 | 0.00 | 0.00 |
| Alive Pred + No prey | 0.25 | 0.60 | 0.15 | 0.00 | 0.00 | 0.00 |
| Alive Prey + Pred | 0.45 | 0.50 | 0.05 | 0.00 | 0.00 | 0.00 |
| Alive Prey + No pred | 0.35 | 0.35 | 0.30 | 0.00 | 0.00 | 0.00 |
| Pregnant + Unborn child | 0.00 | 0.00 | 0.00 | 1.00 | 0.00 | 0.00 |
| Dead | 0.00 | 0.00 | 0.00 | 0.00 | 1.00 | 0.00 |
| Unborn + Npp* | 0.00 | 0.00 | 0.00 | 0.00 | 0.00 | 1.00 |

Table 7: Transition matrix $\Psi_3$

|  | Die | Move | Turn pregnant | Turn alive | Stay dead | Stay unborn |
|---|---|---|---|---|---|---|
| Alive Pred + Prey | 0.15 | 0.30 | 0.55 | 0.00 | 0.00 | 0.00 |
| Alive Pred + No prey | 0.30 | 0.55 | 0.15 | 0.00 | 0.00 | 0.00 |
| Alive Prey + Pred | 0.70 | 0.20 | 0.10 | 0.00 | 0.00 | 0.00 |
| Alive Prey + No pred | 0.10 | 0.40 | 0.50 | 0.00 | 0.00 | 0.00 |
| Pregnant + Unborn child | 0.00 | 0.00 | 0.00 | 1.00 | 0.00 | 0.00 |
| Dead | 0.00 | 0.00 | 0.00 | 0.00 | 1.00 | 0.00 |
| Unborn + Npp* | 0.00 | 0.00 | 0.00 | 0.00 | 0.00 | 1.00 |

Table 8: Transition matrix $\Psi_4$

|  | Die | Move | Turn pregnant | Turn alive | Stay dead | Stay unborn |
|---|---|---|---|---|---|---|
| Alive Pred + Prey | 0.15 | 0.35 | 0.50 | 0.00 | 0.00 | 0.00 |
| Alive Pred + No prey | 0.25 | 0.45 | 0.30 | 0.00 | 0.00 | 0.00 |
| Alive Prey + Pred | 0.45 | 0.40 | 0.15 | 0.00 | 0.00 | 0.00 |
| Alive Prey + No pred | 0.30 | 0.40 | 0.30 | 0.00 | 0.00 | 0.00 |
| Pregnant + Unborn child | 0.00 | 0.00 | 0.00 | 1.00 | 0.00 | 0.00 |
| Dead | 0.00 | 0.00 | 0.00 | 0.00 | 1.00 | 0.00 |
| Unborn + Npp* | 0.00 | 0.00 | 0.00 | 0.00 | 0.00 | 1.00 |

* Not pregnant parent.

**Algorithm 3** Predator-Prey model

---

**Require:** agent set $A = \{1, 2, \ldots, n\}$, grid size $L$, transition matrix $\Psi$, max steps $T$

1: **for all** $i \in A$ **do**
2:      $K^{(i)} \sim \mathrm{Uniform}(\{\mathit{Prey}, \mathit{Predator}\})$                 $\triangleright$ Assign kind
3:      $f_0^{(i)} \sim \mathrm{Uniform}(\{\mathit{Alive}, \mathit{Unborn}\})$           $\triangleright$ Initial phase
4:      **if** $f_0^{(i)} = \mathit{Unborn}$ **then**
5:          $\mathrm{parent}(i) \sim \mathrm{Uniform}(\{j \in A : j \neq i \wedge K^{(j)} = K^{(i)}\})$    $\triangleright$ Assign parent
6:      **end if**
7:      $\mathbf{Z}_0^{(i)} = (K^{(i)}, f_0^{(i)}, x_0^{(i)}, y_0^{(i)})$                $\triangleright$ Agent state
8: **end for**
9: **for** $t = 0, \ldots, T - 1$ **do**
10:      **for all** $i \in A$ **do**
11:          **switch** $f_t^{(i)}$
12:              **case** $\mathit{Alive}$                $\triangleright$ Alive agent dynamics
13:                  $\mathcal{N}_t^{(i)} \leftarrow \mathrm{VonNeumann}(x_t^{(i)}, y_t^{(i)})$    $\triangleright$ Von Neumann neighborhood
14:                  $\mathit{neighbor} \leftarrow \exists j \in A : (x_t^{(j)}, y_t^{(j)}) \in \mathcal{N}_t^{(i)} \wedge K^{(j)} \neq K^{(i)}$ $\triangleright$ Find opposite-kind neighbors
15:                  $f_{t+1}^{(i)} \sim \mathrm{Categorical}\big(\Psi(K^{(i)}, f_t^{(i)}, \mathit{neighbor})\big)$    $\triangleright$ Random life phase update
16:                  **if** $f_{t+1}^{(i)} = \mathit{Alive}$ **then**
17:                      $(x_{t+1}^{(i)}, y_{t+1}^{(i)}) \sim \mathrm{Uniform}(\mathcal{N}_t^{(i)})$      $\triangleright$ Move randomly
18:                  **else if** $f_{t+1}^{(i)} = \mathit{Dead}$ **then**
19:                      $(x_{t+1}^{(i)}, y_{t+1}^{(i)}) \leftarrow (\emptyset, \emptyset)$          $\triangleright$ Agent dies
20:                  **else if** $f_{t+1}^{(i)} = \mathit{Pregnant}$ **then**
21:                      $(x_{t+1}^{(i)}, y_{t+1}^{(i)}) \leftarrow (x_t^{(i)}, y_t^{(i)})$        $\triangleright$ Stay in place
22:                  **end if**
23:              **case** $\mathit{Pregnant}$                   $\triangleright$ Birth
24:                  $f_{t+1}^{(i)} \leftarrow \mathit{Alive}$
25:                  $(x_{t+1}^{(i)}, y_{t+1}^{(i)}) \leftarrow (x_t^{(i)}, y_t^{(i)})$        $\triangleright$ Stay in place
26:              **case** $\mathit{Dead}$
27:                  $f_{t+1}^{(i)} \leftarrow \mathit{Dead}$              $\triangleright$ Remain dead
28:                  $(x_{t+1}^{(i)}, y_{t+1}^{(i)}) \leftarrow (\emptyset, \emptyset)$
29:              **case** $\mathit{Unborn}$
30:                  $j \leftarrow \mathrm{parent}(i)$                $\triangleright$ Get parent
31:                  **if** $f_t^{(j)} = \mathit{Pregnant}$ **then**
32:                      $f_{t+1}^{(i)} \leftarrow \mathit{Alive}$        $\triangleright$ Born if parent is pregnant
33:                      $\mathcal{N}_t^{(j)} \leftarrow \mathrm{VonNeumann}(x_t^{(j)}, y_t^{(j)})$
34:                      $(x_{t+1}^{(i)}, y_{t+1}^{(i)}) \sim \mathrm{Uniform}(\mathcal{N}_t^{(j)})$    $\triangleright$ Place near parent
35:                  **else**
36:                      $f_{t+1}^{(i)} \leftarrow \mathit{Unborn}$          $\triangleright$ Remain unborn
37:                      $(x_{t+1}^{(i)}, y_{t+1}^{(i)}) \leftarrow (\emptyset, \emptyset)$
38:                  **end if**
39:      **end for**
40:      **if** $\forall a \in A,\ f_t^{(a)} \in \{\mathit{Dead}, \mathit{Unborn}\}$ **then**
41:          **break**                    $\triangleright$ Simulation ends
42:      **end if**
43: **end for**

---

## C  Further experimental details and results

In this section, we provide additional details on the experiments and present further results. In the subsection on experimental design (C.1), we describe the micro and macro metrics used, including the number of points over which these metrics are computed. We then present additional qualitative results on reproducing emergent segregation in the Schelling model (C.2) and emergent oscillations in predator-prey dynamics (C.3). Then, we provide further quantitative results for both models (C.4). Finally, we provide a discussion on a macro-level baseline (C.5), specifically an autoregressive model of order one, or AR(1), a standard time-series model.

### C.1  Experimental design

**Experiment details**   In our experiments, we considered three combinations of the parameter $\xi$ for the Schelling ABM and four combinations of the matrix $\Psi$ for the Predator-Prey ABM. For each parameter setting, we generated 8 training datasets obtained with different initial seeds and trained a surrogate model and two ablated models for each. In total, we trained 168 models, 96 ($8 \times 4 \times 3$) for Predator-Prey, and 72 ($8 \times 3 \times 3$) for Schelling. All our evaluations are done across these 8 models per parameter configuration.

We fixed some of the ABM parameters across experiments, which are reported in Table 9. For Predator-Prey, density refers to the density of agents that are initialized as *Alive*, whereas the number of agents refers to the total number of agents in the system (*Alive*, *Dead*, *Pregnant*, and *Unborn* agents). *Color Ratio* indicates the ratio between the number of black and white agents; *Kind Ratio* indicates the ratio between the number of predators and the number of preys.

Table 9: ABM parameters in our experiments

| Component | Schelling | Predator-Prey |
|---|---|---|
| Grid size | $51 \times 51$ | $32 \times 32$ |
| Density | 0.75 | 0.3 |
| Agents | 1950 | 2048 |
| Color/Kind Ratio | 1:1 | 1:1 |

**Micro evaluation.** We evaluate how well the surrogate captures individual behavior of agents on a *future* ramification dataset of $T = 25$ timesteps. We generate this *out-of-training* dataset by giving as initial condition the last system configuration $\mathbf{Z}_{T-1}[r = 0]$ from the training ramification dataset. Thus, for each agent $i \in A$ we have 24 initial conditions $(\mathbf{Z}_t^{(i)}, \{\mathbf{Z}_t^{(j)}\}_{j \in N_t^{(i)}})$ and 500 outcomes $\mathbf{Z}_{t+1}^{(i)}$ to build 24 ground truth probability distributions, such as equation (1). Then, we use our surrogate model to generate 500 outcomes $\mathbf{Z}_{t+1}^{(i)}$ given as condition $(\mathbf{Z}_t^{(i)}, \{\mathbf{Z}_t^{(j)}\}_{j \in N_t^{(i)}})$, producing 24 probability distributions such as (1) for each agent $i \in A$, which we compare to the probability distributions from the ground truth. For Schelling, we measure the EMD on the marginals of the coordinates $x$ and $y$. For Predator-Prey, we measure the EMD on the distributions of the phases $f_t^{(i)}$, and fix the distance between the different phases to 1, since they represent a categorical variable. For each of the 8 experiments, we calculate the mean EMD over all agents and all timesteps. For Schelling, we evaluate the EMD on 1950 distributions (one per agent) for the $x$ coordinate and 1950 distributions for the $y$ coordinate, over 24 timesteps, yielding 93600 EMD entries. For Predator-Prey, we evaluate the EMD on 2048 distributions (one per agent) over 24 timesteps, yielding 49152 EMD entries. The box plots in Figure 4 have as entries the 8 mean EMD values for the surrogate, 8 mean EMD values for the diffusion-only ablated model, and 8 mean EMD values for the GNN-only ablated model.

**Macro evaluation.** We evaluate how well the surrogate model reproduces system-level behavior by tracking summary statistics over time. We generate 100 independent simulations of 25 timesteps beyond the training horizon for each model, and compute the symmetric mean absolute error (sMAPE) between the mean ground-truth trajectory and the mean surrogate-generated trajectory across these 100 independent simulations. In the case of Schelling, where we only track the number of *happy* agents, the sMAPE definition is straightforward. Let $A_t$ and $F_t$ be the mean number of happy agents

across the 100 independent simulations from the ground-truth and the surrogate model respectively. Then, the sMAPE is computed as:

$$sMAPE_{\text{schelling}} = \frac{2}{T} \sum_{t=1}^{T} \frac{|A_t - F_t|}{|A_t| + |F_t|} \tag{6}$$

For Predator-Prey, we track the number of predators and preys on the grid, which follow two distinct trajectories. Thus, for each *kind* we apply Formula 6 separately and get $sMAPE_{preys}$ and $sMAPE_{predators}$. Then, to work with a single value, we calculate the mean value:

$$sMAPE_{\text{predprey}} = \frac{1}{2}(sMAPE_{preys} + sMAPE_{predators}) \tag{7}$$

For each of the 8 experiments, we calculate the sMAPE over 25 timesteps and then compute its mean. The box plots in Figure 4 have as entries the 8 mean sMAPE values for the surrogate, 8 mean sMAPE values for the diffusion-only ablated model, and 8 mean sMAPE values for the GNN-only ablated model.

## C.2   Reproducing emergent segregation

Figure 2 in the main text illustrated how, for three selected time steps ($t = 0$, $t = 15$, and $t = 30$) and three different values of the intolerance threshold $\xi$, our surrogate model successfully reproduced the qualitative dynamics of the original ABM. In contrast, the ablated models failed to capture these dynamics.

Figures 8, 9, and 10 provide a more detailed view of the system's evolution for the values $\xi_1$, $\xi_2$, and $\xi_3$, respectively. In addition to the previously shown snapshots, these supplementary figures include intermediate time steps ($t = 5$, $t = 10$, $t = 20$, and $t = 25$), offering a more complete picture of the dynamics.

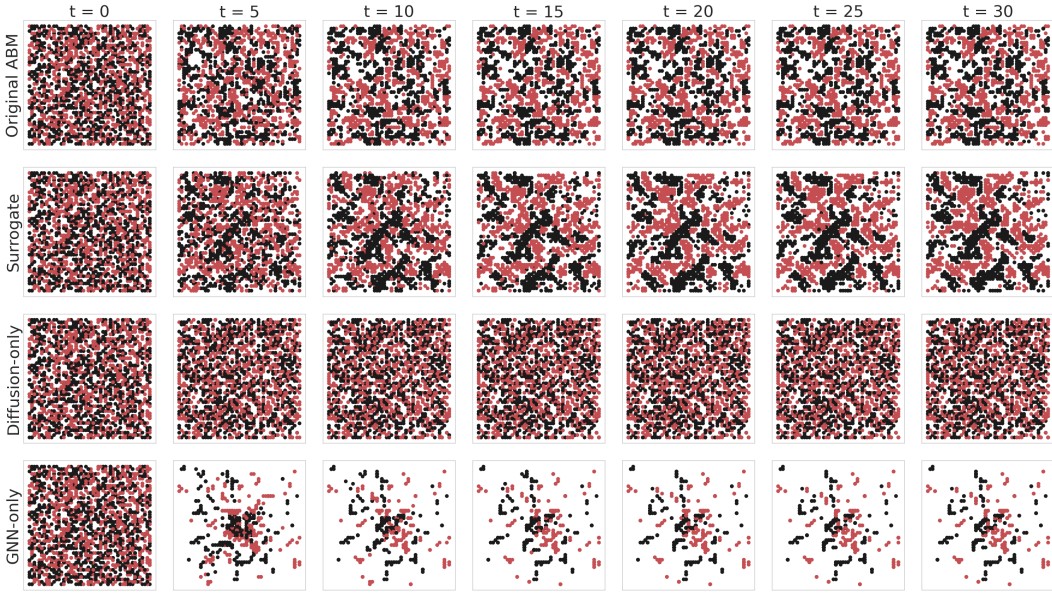

Figure 8: Evolution of the position of black and red agents in the Schelling model, for tolerance thresholds $\xi = \xi_1 = 0.625$. We compare the ground truth (top row) with our surrogate (second row) and the ablations (third and fourth row).

Figure 8 illustrates that the original ABM rapidly evolves toward a segregated configuration, which becomes visible as early as $t = 5$. However, the resulting clusters remain relatively small, indicating that a low intolerance threshold allows for a moderate level of social mixing. This motivates the label "Low Segregation" for the $\xi_1$ parameter setting in Figure 4. The surrogate model accurately reproduces

these qualitative patterns, albeit with slightly larger clusters. In contrast, the diffusion-only ablation fails to capture the emergent spatial structure, remaining close to the initial random configuration across time steps. Instead, the GNN-only ablation ends up overlapping most agents on the same coordinates.

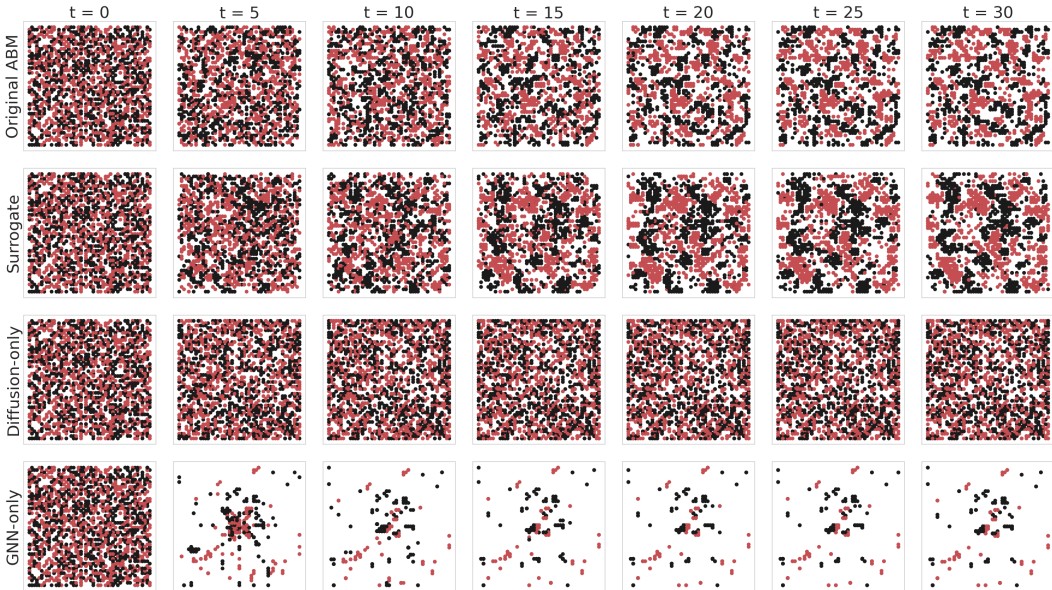

Figure 9: Evolution of the position of black and red agents in the Schelling model, for tolerance thresholds $\xi = \xi_2 = 0.75$. We compare the ground truth (top row) with our surrogate (second row) and the ablations (third and fourth row).

Figure 9 shows that, for $\xi = \xi_2$, the original ABM also converges toward a segregated state, but the convergence occurs more slowly than in the $\xi_1$ case. The resulting clusters are larger and more distinct, reflecting a "High Segregation" scenario. The surrogate model again successfully replicates these dynamics, while the ablation models continue to exhibit no structured behavior.

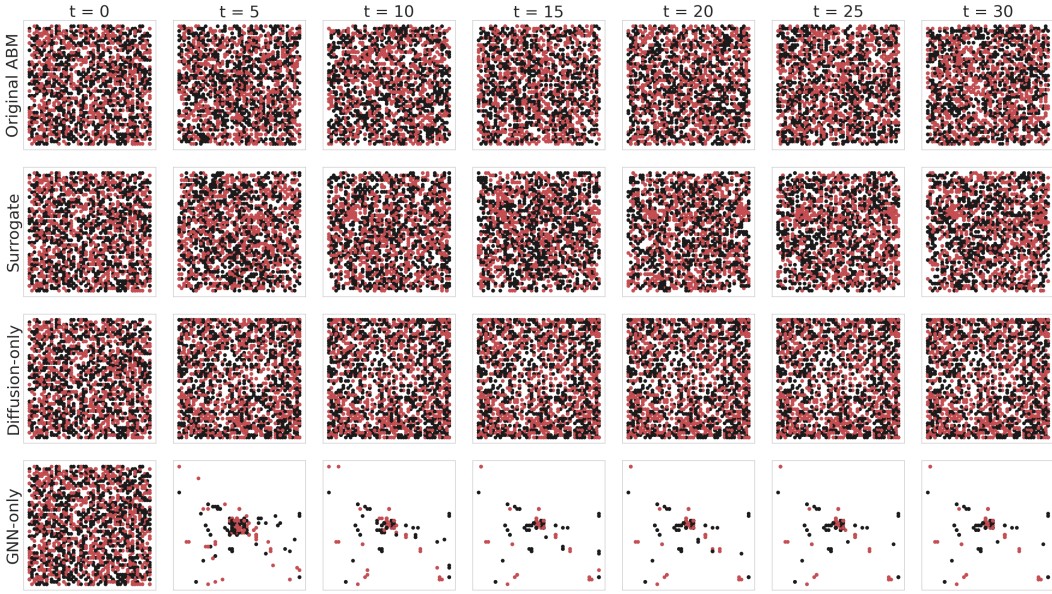

Figure 10: Evolution of the position of black and red agents in the Schelling model, for tolerance thresholds $\xi = \xi_3 = 0.875$. We compare the ground truth (top row) with our surrogate (second row) and the ablations (third and fourth row).

Finally, Figure 10 demonstrates that, for $\xi = \xi_3$, the original ABM does not converge to a stable configuration. Instead, the high intolerance threshold causes agents to continuously relocate, preventing the emergence of segregated clusters. In this degenerate case, both the surrogate and the diffusion-only ablation model correctly reproduce the persistent disorder of the original dynamics, while the GNN-only ablation still fails.

## C.3 Reproducing emergent oscillations in predator-prey dynamics

Figure 3 in the main text compares population trajectories from the ground-truth Predator-Prey ABM, our surrogate model, and the ablated model. For both parameter sets shown ($\Psi_1$ and $\Psi_4$), the surrogate accurately reproduces the stochastic dynamics beyond the training window, while the ablation fails to capture the key oscillation patterns.

Figure 11 extends the previous analysis to parameter sets $\Psi_2$ and $\Psi_3$, which produce, respectively, a monotonic decline in prey and predator populations, and oscillatory dynamics for preys but not for predators. The case of $\Psi_2$ is particularly illustrative: despite its apparent simplicity, the ablated models fail to reproduce the monotonic trends, showing an unrealistic spike in both populations immediately after the training window in the diffusion-only ablated model, and a roughly constant number of preys and predators in the GNN-only ablated model. In contrast, the surrogate model accurately captures the expected decay. Under $\Psi_3$, the surrogate successfully replicates the oscillations in the prey population and the stable predator trend, while the diffusion-only ablation once again outputs generic dynamics, largely insensitive to the underlying regime, and the GNN-only generates an oscillating dynamic for both. This highlights the ablation models' inability to distinguish between qualitatively different behaviors.

Beyond aggregate population counts, it is instructive to examine the spatial distribution of predators and preys over time. Prior studies [45, 46] have shown that similar predator-prey models give rise to rich spatial dynamics, characterized by the spontaneous emergence of structured patterns (see, e.g., Figure 1.3 in [47]). Starting from initially random configurations, the interactions between agents give rise to both short- and long-range spatial correlations, with predators and preys organizing into dynamic clusters and propagating waves. These patterns are reminiscent of those observed in spatially extended reaction-diffusion systems and bear a strong resemblance to spatial chaos phenomena in evolutionary game theory [27].

We observe similar spatial patterns in our predator-prey ABM. Figures 12, 13, 14, and 15 show the evolution of the positions of predators and preys on the grid.

Figure 12 illustrates the spatial dynamics under parameter set $\Psi = \Psi_1$, which is characterized by oscillations in predator and prey populations. The ground-truth model displays a rise in population densities mid-simulation, followed by a near-extinction phase toward the end, in line with the temporal trends shown in Figure 3. Notably, the ground-truth also exhibits distinct spatial patterns, with predators and preys forming dynamic clusters. The surrogate model successfully reproduces both the population dynamics and the emergent spatial structures, whereas the ablated models fail to capture any meaningful spatial organization.

Similar results are observed in Figures 13, 14, and 15, which capture different dynamic regimes for predators and preys. In all cases, the surrogate model accurately reproduces both spatial and temporal patterns, while the ablated models consistently fail to capture the underlying dynamics or spatial structure.

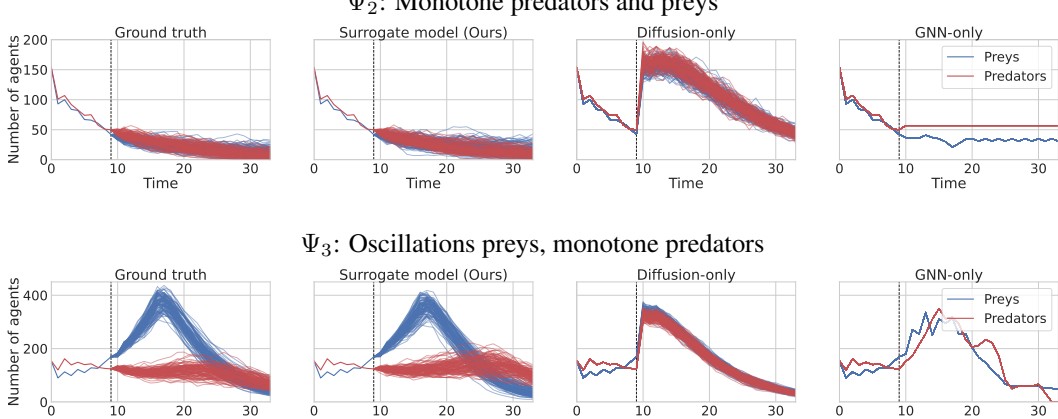

Figure 11: Forecasting macro-level summary statistics (here, the number of alive preys and predators over time), starting from the last condition seen in training, for 100 independent simulation runs, under configuration $\Psi_2$ (monotonic dynamics for both predators and preys, top) and $\Psi_3$ (oscillations only for predators, bottom). First colummn: original ABM simulations. Second column: surrogate. Third and fourth column: ablations. The dashed vertical line indicates the end of the training phase for the surrogate and ablation models.

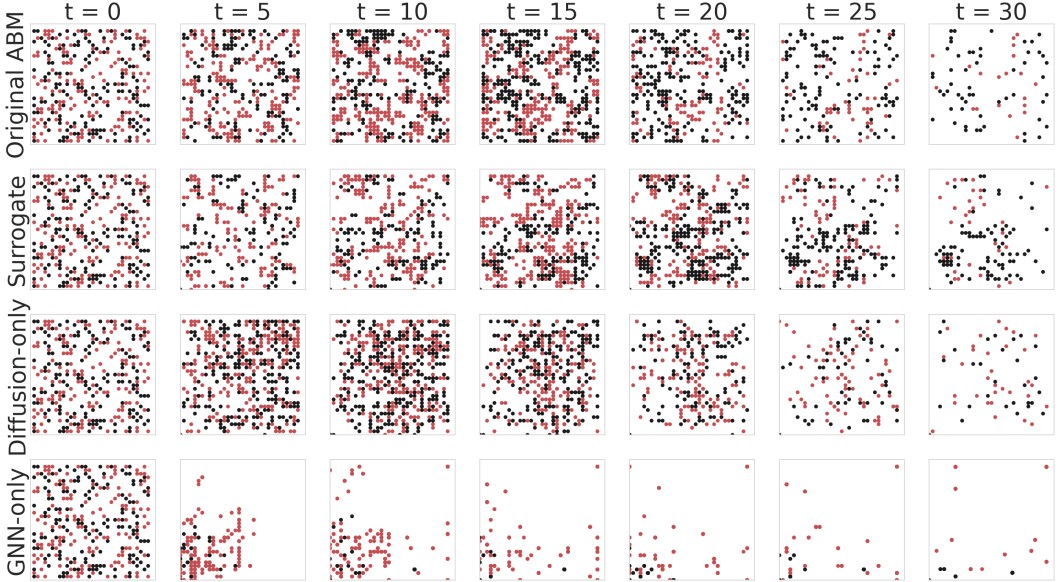

Figure 12: Evolution of the position of preys (black) and predators (red) in the predator-prey ABM, for parameters $\Psi = \Psi_1$. We compare the ground truth (top row) with our surrogate (second row) and the ablations (third and fourth row).

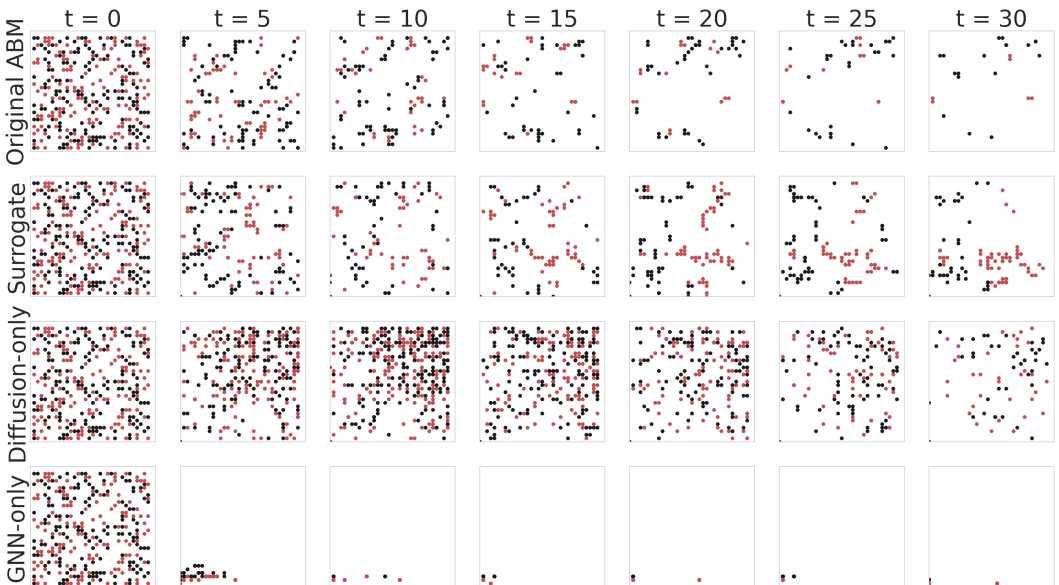

Figure 13: Evolution of the position of preys (black) and predators (red) in the predator-prey ABM, for parameters $\Psi = \Psi_2$. We compare the ground truth (top row) with our surrogate (second row) and the ablations (third and fourth row).

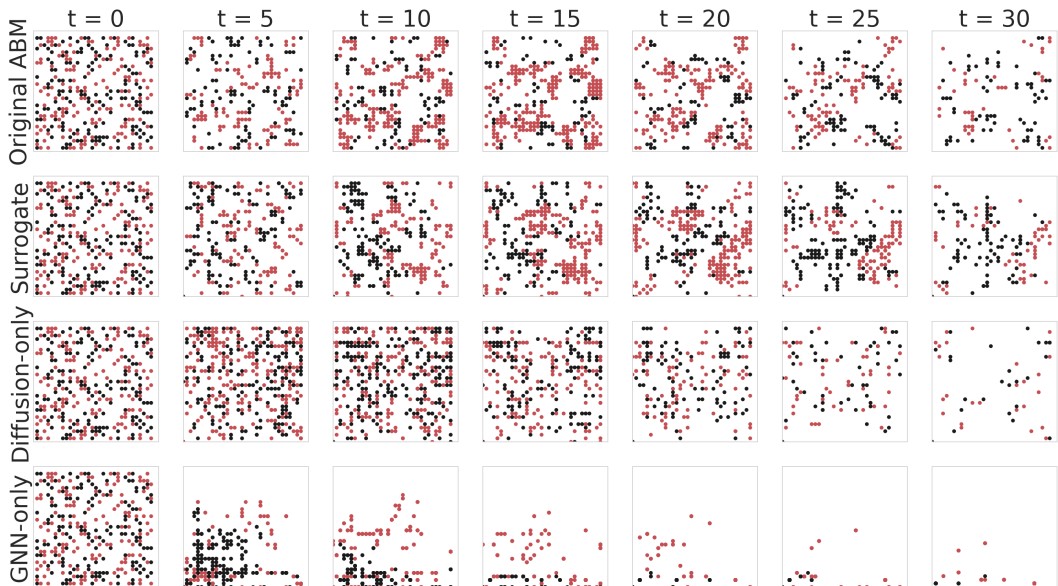

Figure 14: Evolution of the position of preys (black) and predators (red) in the predator-prey ABM, for parameters $\Psi = \Psi_3$. We compare the ground truth (top row) with our surrogate (second row) and the ablations (third and fourth row).

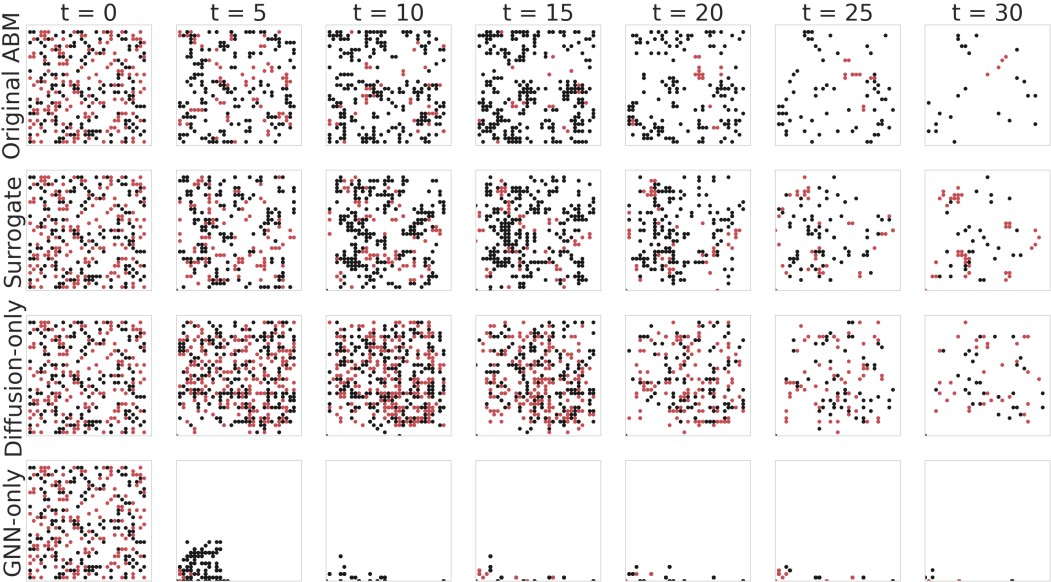

Figure 15: Evolution of the position of preys (black) and predators (red) in the predator-prey ABM, for parameters $\Psi = \Psi_4$. We compare the ground truth (top row) with our surrogate (second row) and the ablations (third and fourth row).

## C.4 Quantitative results

In this section, we present additional details regarding the quantitative evaluation discussed in Section 4.2 (page 8). In particular here we detail further the micro-level evaluation; that is, how much our surrogate model can reproduce the distribution of possible states of an agent at $t$ (that is, $\mathbf{Z}_t^{(i)}$) given its past state $\mathbf{Z}_{t-1}^{(i)}$. To better understand how this comparison works, Figure 16 (left side) shows the distribution for a single agent of the position $x_t^{(i)}$, one of the components of the state $\mathbf{Z}_t^{(i)}$, for each of the two possible past conditions in the Schelling model, *happy* and *unhappy*. We see that when the agent is *happy*, its position remains fixed, while in the case the agent is *unhappy*, it moves randomly with a certain distribution peaked around the starting point (implicitly defined by the ABM, see Algorithm 2). Our surrogate model aims at reproducing this distribution, without knowing the original ABM and without access to the latent variable *happy/unhappy*, but only observing the sequence of states and the graph of interactions. The distributions obtained by our surrogate in the same starting conditions are shown on the right side of the figure. To evaluate the quality of this reconstruction, we quantify it as the Earth Mover's Distance between the distributions, for each agent in each timestep, and then aggregate these measures.

The result of this comparison is shown in Figure 4 of the main text. Here, we present additional experiments showing how these results change across different conditions.

**Schelling model.** Figure 17 shows the distribution of the EMD scores obtained by our surrogate model and by the ablation model only for the stochastic rules of the Schelling model. In this ABM, the stochasticity lies in the random movement of the *unhappy* agents (Algorithm 2, lines 16-25). We see that our surrogate can capture these distributions well (EMD averages below 5, considering that the scale is given by the length of the grid, $L = 50$) and better than the surrogate model.

**Predator-prey model.** Figure 18 instead shows the distribution of the EMD scores only for the stochastic rules of the Predator-Prey model. Here, the stochasticity lies in the *Alive* phase, where the agent might transition to another phase depending on its interaction with its neighbors. We see that also in this model our surrogate is able to capture the stochasticity well, with EMD averaging below 0.1 for each of the four configurations $\Psi_{1-4}$. By contrast, the diffusion-only ablation model's EMD for these same stochastic rules averages above 0.12, up to values around 0.2. Here, the scale is in $[0, 1]$ since this error is measured on the life phase binary vector. We further investigate these results in Figure 19 by dividing them by agent state in each configuration. Here, the stochastic transitions happen only in the first column (*Alive*). This figure confirms that the EMD values are quite low, and lower than those obtained by the ablation model. Moreover, here we observe that also the deterministic transitions are recovered with precision (error is always below $10^{-3}$)) by our surrogate model.

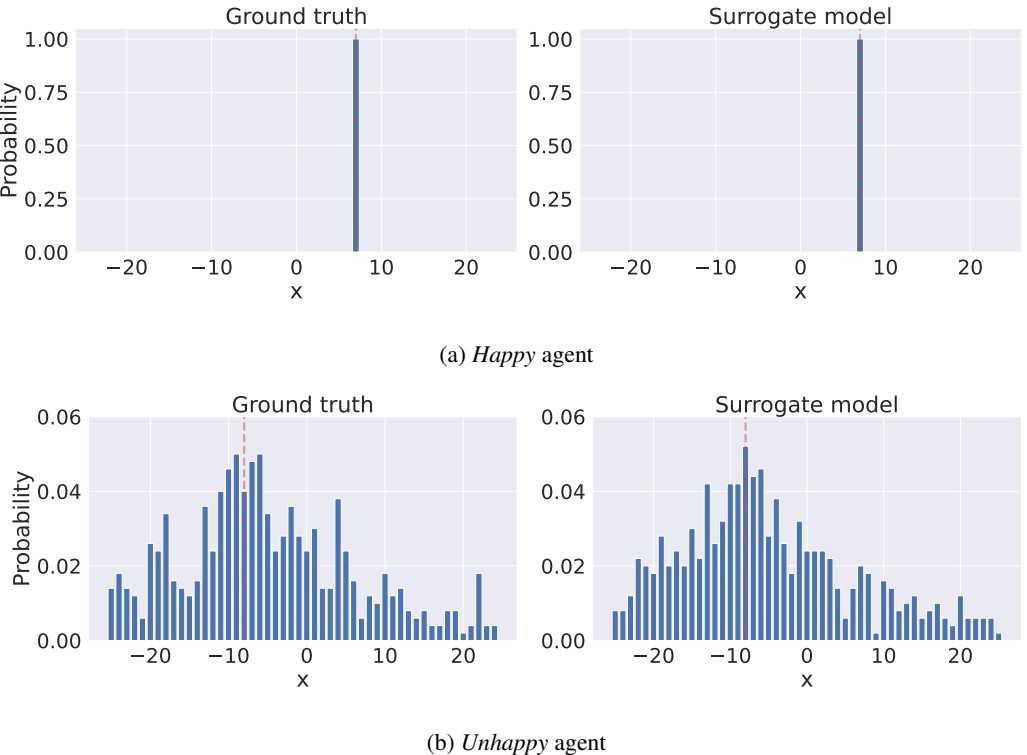

(a) *Happy* agent

(b) *Unhappy* agent

Figure 16: Example distributions of the position of a single agent in the Schelling model, in two different conditions: (a) *happy* agent; (b) *unhappy* agent. Histograms on the left represent the original ground-truth ABM, those on the right the ones obtained by our surrogate model. The dashed red vertical line indicates the initial position of the agent. Note that the coordinates have been rescaled to [-25, 25], compared to the [0, 50] interval used for ease of exposition in Algorithm 2.

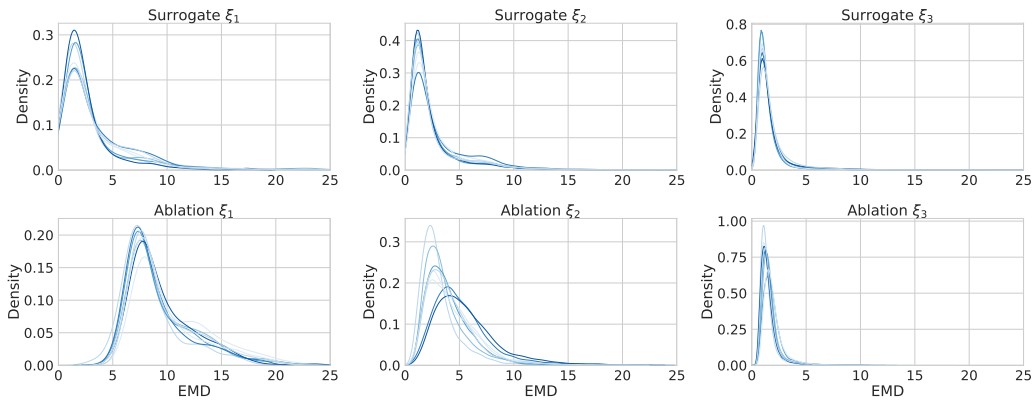

Figure 17: Distribution of errors (EMD) obtained by our surrogate model (top row) and by the diffusion-only ablation model (bottom row) only for the stochastic rules of the Schelling ABM for each considered configuration of the parameter $\xi$ of the ABM. Different colors indicate independent experiments that only differ by the random seed used to generate the ground truth data.

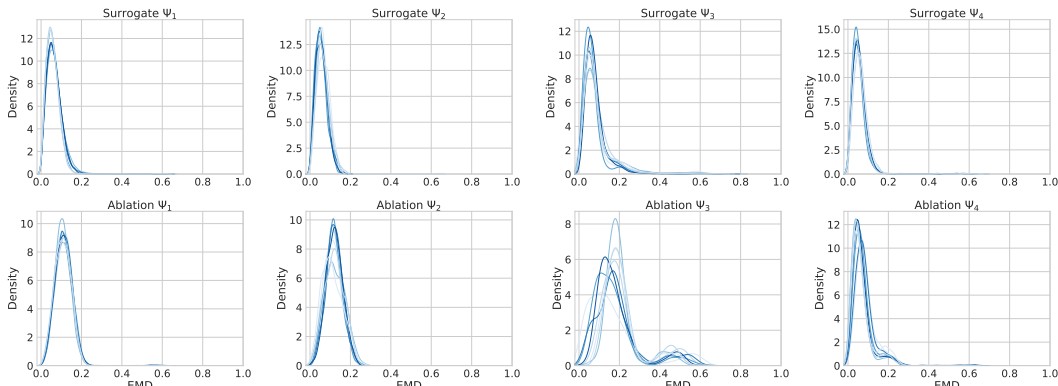

Figure 18: Distribution of errors (EMD) obtained by our surrogate model (top row) and by the diffusion-only ablation model (bottom row) only for the stochastic rules of the Predator-Prey ABM for each considered configuration of the parameter matrix $\Psi$ of the ABM. Different shades indicate independent experiments that only differ by the random seed used to generate the ground truth data.

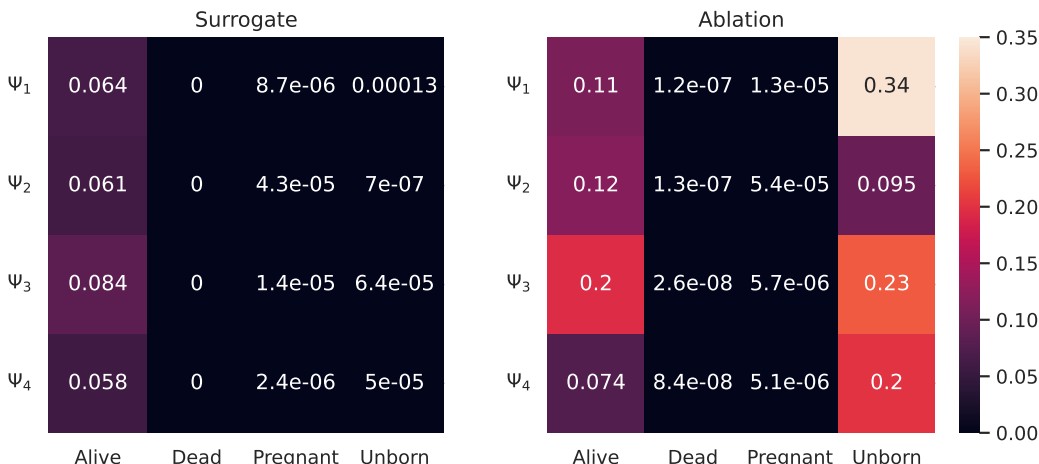

Figure 19: Mean error (EMD) obtained by our surrogate (on the left) and by the diffusion-only ablation model (on the right) for each configuration of the Predator-Prey ABM (on the rows, $\Psi_{1-4}$) and for each initial state of an agent (on the columns).

### C.5   Macro baseline (AR1)

In this subsection, we show that a simple yet commonly used time-series baseline — the autoregressive model of order one (AR(1)) — completely fails to reproduce the macro-level dynamics of our system. We focus on the predator–prey model, although qualitatively similar results hold for the Schelling model and for higher-order time-series models (e.g., AR(2)).

The AR(1) model assumes that the value of a time series at step $t$ depends linearly on its previous value:

$$x_t = \phi x_{t-1} + \epsilon_t,$$

where $\phi$ is the autoregressive coefficient and $\epsilon_t \sim \mathcal{N}(0, \sigma^2)$ is Gaussian noise. We fit the model separately to each macro variable using least squares estimation on the training portion of the simulated data. The fitted coefficients $\hat{\phi}$ and $\hat{\sigma}$ are then used to generate out-of-sample forecasts, producing trajectories that can be directly compared with those generated by our surrogate model.

As shown in Figure 20, the AR(1) baseline simply extrapolates the last observed trend in the training data, completely failing to reproduce the non-monotonic, oscillatory patterns characteristic of the underlying dynamics — and even performing poorly in capturing monotonic trends. This limitation arises because the system's behavior is defined at the micro level, while macro-level variables are

only aggregate summaries of those micro interactions. Consequently, a surrogate model that learns from micro-level states, as ours does, is inherently better positioned to capture both the micro and the emergent macro dynamics.

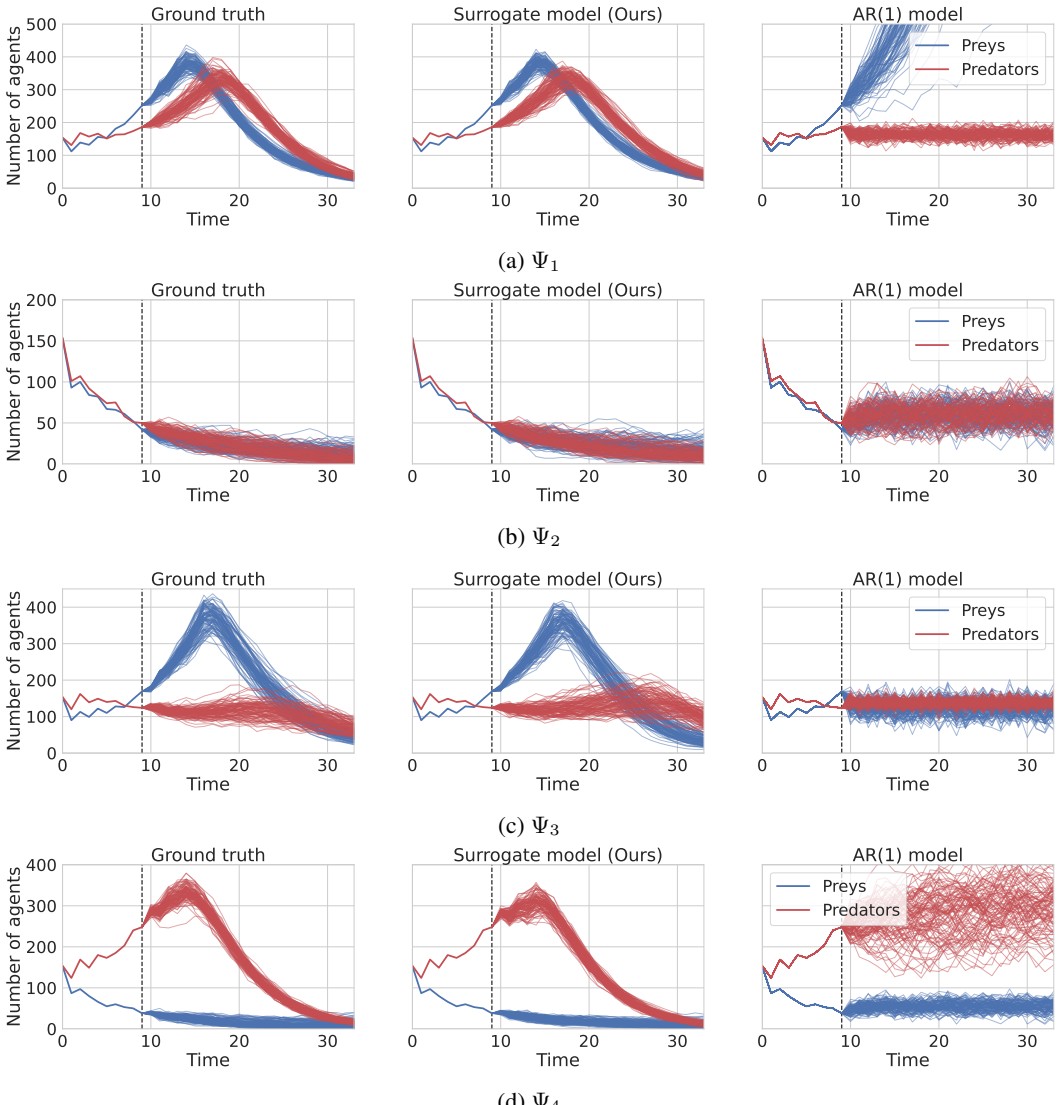

(a) $\Psi_1$

(b) $\Psi_2$

(c) $\Psi_3$

(d) $\Psi_4$

Figure 20: Comparison of $\Psi_1$–$\Psi_4$ under the surrogate GDN model and an AR(1) baseline.

# SM References

[45] André M De Roos, Edward McCauley, and William G Wilson. Mobility versus density-limited predator-prey dynamics on different spatial scales. *Proceedings of the Royal Society of London. Series B: Biological Sciences*, 246(1316):117–122, 1991.

[46] Douglas D Donalson and Roger M Nisbet. Population dynamics and spatial scale: effects of system size on population persistence. *Ecology*, 80(8):2492–2507, 1999.

[47] Volker Grimm and Steven F Railsback. Individual-based modeling and ecology. In *Individual-based modeling and ecology*. Princeton university press, 2013.

[48] Alexander Quinn Nichol and Prafulla Dhariwal. Improved denoising diffusion probabilistic models. In *Proceedings of the 38th International Conference on Machine Learning*, volume 139, pages 8162–8171. PMLR, 2021.

