# OpenReview forum: "Learning Individual Behavior in Agent-Based Models with Graph Diffusion Networks"
_NeurIPS.cc/2025/Conference — NeurIPS 2025 poster_

### Official Review · Reviewer_yYgi · 2025-06-16

**Clarity:** 2
**Significance:** 2
**Originality:** 3
**Rating:** 4
**Confidence:** 1

**Summary:**

This paper proposes a method to predict the behavior of Agent-Based Models (ABMs). Specifically, the authors trained a differentiable surrogate of an ABM, which is a GNN, using ABM-generated data.

**Questions:**

1. Please take a look at the Weaknesses point 1. One unclear description example is:
(L30) However, this progress is occurring despite the absence of principled methods to systematically align ABMs with real-world data. While various approaches have been proposed for calibrating macro-level parameters of ABMs [29], there are still no established methods for tuning the micro-level behaviors of individual agents to data.
Q: Why does "align ABMs with real-world data" relate to "tuning the micro-level behaviors of individual agents to data"? Doesn't "calibrating macro-level parameters of ABMs" belong to "systematically align ABMs with real-world data"? Also, this paper also doesn't use real-world data, so what is the advantage of the proposed method compared with previous ones?

2. Could the authors provide the related works and also add some baselines for experimental comparison?

**Ethical Concerns:**

["NO or VERY MINOR ethics concerns only"]

**Final Justification:**

The authors directly and clearly answer my questions. I'm very unfamiliar with the area of ABM, so I can only review from a general machine learning and GNN perspective.

**Limitations:**

Yes. There are limitation discussions in the Section 5.

**Quality:**

3

**Strengths And Weaknesses:**

Strengths
This paper explained the experimental details very clearly.

Weaknesses

1. The paper is clear in how the experiments are conducted, however, it is unclear in the motivation, related works and baseline selection.
My suggestion is that, the authors can pay more attention to explain 1/ why do we need to model individual agent behaviors, what the benefit is. 2/ explanation of terminologies used to make the paper self-contained, e.g., macro-level and micro-level in the context of Agent-Based Models. 3/ the evidence that the selection of Schelling and predator-prey models are representative enough. 4/ systematic comparison with previous works.

2. In this paper, all the baselines are variants of the proposed models. Also, the authors do not mention the well-recognized baselines in this area, and how previous works are evaluated.

---

> ### Author Rebuttal · Authors · 2025-07-31
>
> We appreciate the reviewer’s detailed and constructive comments. We reply by numbering the weaknesses by W1, W2, … and the questions by Q1, Q2, …, grouping weaknesses and questions where relevant. For each reply, we start with a one-sentence TL;DR and then develop our argument.
>
> ---
> # W1, Q1: Clarity
> **TL;DR**: We clarify our terminology and explain why the predator-prey and Schelling models are representative of ABMs.
>
> We realize that the benefits of **modeling individual behaviors** were not adequately explained.
>
> ABMs are based on defining rules at the level of individual agents – a perspective known as the micro-level. These rules deal only with the characteristics and behaviors of single agents, such as their type (e.g., predator or prey) or positions. Micro-level dynamics are governed by these rules, that act only on these individual features, such as updating an agent’s position over time. The purpose of ABMs is often to observe how these simple micro-level rules give rise to complex, emergent properties at the system-wide level, referred to as the macro-level. Examples of macro-level outcomes include summary statistics, such as the total number of surviving agents or the total number of agents that are happy of their neighbors, associated to segregation patterns.
>
>
> With this background, we can now explain the value of modeling individual behaviors directly. In fact, we constrain our surrogate to model only micro-level rules. This ensures that the surrogate remains structurally faithful to the original ABM and does not rely on shortcuts such as directly predicting macro-level outcomes. To illustrate this, consider the predator-prey model. A macro-level surrogate might emulate the system using equations like the Lotka–Volterra model, which predicts population counts over time without modeling individual agents. In contrast, our surrogate expresses rules in terms of agent-level attributes, such as the type and position of each agent. This approach leads to two major benefits. First, by modeling only micro-level rules, the surrogate stays aligned with the logic of the original ABM. Second, the surrogate is expressed directly in terms of micro-level variables, allowing for real-data alignment with respect to individual agent attributes – something standard surrogates cannot do.
>
> Traditional surrogates, in fact, can only perform macro-level calibration (e.g., tuning a global threshold like $\epsilon$ in the Schelling model). Our model, however, is differentiable with respect to micro-level inputs, such as the type or position of each individual agent. Thanks to this property, our approach offers practical advantages in integration with real-world data. For instance, when micro-level data is available for a subset of agents (e.g., demographic or spatial attributes), our surrogate can use this information to simulate possible future states.
>
> Therefore, in response to Q1, to faithfully "align ABMs with real-world data," we mean to align the micro-level properties of individual agents to match observed data. Calibrating macro-level parameters only allows to calibrate some key macro-level quantities of the ABM, but its agents remains totally disconnected from any entity observed in the real-world. Consider again the predator-prey model: if we have a real-world data source enumerating some real-world animal positions, aligning this data with the ABM at the micro-level would mean associating one of such real positions to a simulated agent, instead of simply tuning a global parameter for the whole system. In this sense, aligning micro-level states with real-world observations remains an open challenge in the ABM literature, yet it is essential for ensuring that ABMs can be reliably applied in real-world contexts. Our model is the first differentiable surrogate capable of expressing learned stochastic rules in terms of such micro-level attributes. As such, it represents a crucial step toward building tools that can calibrate the micro-level components of ABMs, an often overlooked but essential aspect of systematically aligning ABMs with real-world data.
>
> Regarding the **representativeness of our choice of ABMs**, both the Schelling model and predator-prey models are widely recognized as canonical examples of agent-based models. They are commonly used to introduce the field and are frequently employed in studies that explore calibration techniques and surrogate modeling.
>
> The Schelling model is often cited as one of the earliest ABMs in the social sciences [1], and has served as the sole testbed for the development of data-driven model extraction methods [2]. Similarly, predator-prey models form the foundation for a broad range of more complex models in ecological research [3, 6], and they have played a key role as testbeds for the development of novel calibration and surrogate techniques. For example, in "Data assimilation with agent-based models using Markov chain sampling" [4], the authors test their method for an MCMC-based sampler of possible trajectories on a predator-prey ABM, very similar to ours. It is their only case study. Likewise, in "The use of surrogate models to analyse agent-based models" [5], one of the three case studies used to compare macro-level surrogates is a predator-prey model. It is worth noting that in all of these works, the predator-prey model is used as a representative of a broader class of ABMs, likely due to its simplicity, intuitive appeal, and ease of explanation to readers across different disciplines.
>
> We will add these considerations and further clarifications to the final version of the manuscript.
>
> [1] Squazzoni, F. (2010). The impact of agent-based models in the social sciences after 15 years of incursion. History of economic ideas: XVIII, 2, 2010, 1000-1037.
>
> [2] Jamali, Ruhollah, Wannes Vermeiren, and Sanja Lazarova-Molnar. "Data-driven agent-based modeling: Experimenting with the Schelling’s model." Procedia Computer Science 238 (2024): 298-305.
>
> [3] McLane, Adam J., et al. "The role of agent-based models in wildlife ecology and management." Ecological modelling 222.8 (2011): 1544-1556.
>
> [4] Tang, Daniel, and Nick Malleson. "Data assimilation with agent-based models using Markov chain sampling." Open Research Europe 2 (2022): 70.
>
> [5] Ten Broeke, Guus, et al. "The use of surrogate models to analyse agent-based models." Journal of Artificial Societies and Social Simulation 24.2 (2021).
>
> [6] Murphy, Kilian J., Simone Ciuti, and Adam Kane. "An introduction to agent‐based models as an accessible surrogate to field‐based research and teaching." Ecology and evolution 10.22 (2020): 12482-12498.
>
> ---
> # W2, Q2: Baselines
> **TL;DR**: Comparisons to baselines is difficult given the novel focus of our approach.
>
> All reviewers raised concerns about the lack of baseline comparisons. We acknowledge this and have expanded our evaluation accordingly. However, our work introduces the first individual-level stochastic surrogate for ABMs—a direction that, to our knowledge, has not been explored before. As such, there are no directly comparable baselines. This is similar to the situation faced by Grattarola et al. (NeurIPS 2021), who introduced the first GNN-based surrogate and were also unable to compare against prior methods.
>
> Below, we summarize our responses to the baseline-related concerns raised by the other reviewers.
>
> Reviewer BiVF suggested we focus on macro-level surrogates. However, most of these surrogates do not model system dynamics directly, but rather map parameters to summary statistics. While a few exceptions exist (e.g., Fadikar et al., Kieu et al.), they still model aggregate variables and struggle to generalize beyond in-sample dynamics. For instance, an AR(1) time-series model fails to reproduce even simple non-monotonic patterns like predator-prey cycles, significantly underperforming compared to our individual-level approach.
>
> In response to reviewer McsJ suggestions, we conducted an additional evaluation, training a GNN-only ablation. While this version performs adequately/decently in deterministic settings, it struggles with stochastic dynamics, showing much higher errors than the full GDN, particularly in the predator-prey model. In the Schelling model, where behavior is more deterministic under certain parameterizations, the GNN-only performance was closer to GDN. Reviewer McsJ also suggested that we considered alternative  GNNs (e.g., Grattarola et al., Gravina et al.), but these alternatives  suffer from the same structural limitation of our ablation study: they predict deterministic outcomes and cannot model stochastic transitions. This is supported by the underperformance of the GNN-only ablation. Another proposed baseline is the work from Casert et al., a promising Transformer-based stochastic emulator. However, their method is not applicable to our setting. It assumes fixed-size state spaces, continuous time, and explicit enumeration of all allowed transitions — constraints incompatible with ABMs like Schelling or Predator-Prey, where state spaces are large and dynamic.
>
> We will include the new experiments, along with the AR(1) baseline and an extended discussion, in the revised paper and supplement.
>
>
> Finally, the reviewer raised a concern regarding the lack of related work, requesting that we “provide the related works.” While we will revise the manuscript to include the expanded discussion presented above, we believe that Section 2 already covers the key contributions of the most relevant prior work in this area. That said, we are fully open to including additional references and discussion if the reviewer can point to specific missing works or directions that merit inclusion.
>
> ---
> We welcome any follow-up questions and would be happy to provide further details or clarifications.

---

> ### Comment · Reviewer_yYgi · 2025-08-05
>
> Thanks for the response. It addresses my questions.

---

### Official Review · Reviewer_McsJ · 2025-06-26

**Clarity:** 3
**Significance:** 2
**Originality:** 3
**Rating:** 5
**Confidence:** 3

**Summary:**

The authors propose a novel framework, the Graph Diffusion Network (GDN), that  combines graph neural networks and diffusion models to learn a fully differentiable surrogate model directly from ABM-generated simulation data. The core innovation is a shift in focus from emulating macro-level system outputs to directly modeling the behavior of individual agents. The framework is validated on two canonical ABMs: Schelling's segregation model and a Predator-Prey ecosystem. The experiments demonstrate that the learned surrogate not only replicates the conditional next-state distributions at the individual agent (micro) level but also accurately reproduces the emergent, system-level (macro) dynamics, even when forecasting beyond the training period. The results show a significant improvement over an ablation variant that does not model the interaction graph, highlighting the importance of the proposed architecture.

**Questions:**

- The paper should include a discussion on the computational costs and scalability of the proposed framework. How the method scales with increasing agent count, time horizon, or stochastic branches per step. A more detailed discussion would be valuable.
- To assess the importance of modeling stochasticity, the authors should include the diffusion-removed version of the model in their experiments. Furthermore, comparisons against other differentiable simulators—such as Transformer-based architectures or temporal GNNs—would provide stronger evidence for the effectiveness of the proposed GDN framework.
- The paper does not clearly state the type of GNN employed (e.g., GraphSAGE, GCN, GAT). Since the choice of message-passing scheme can significantly impact expressiveness and inductive bias, the authors should clarify the GNN variant used and its implications for learning agent interactions.

**Ethical Concerns:**

["NO or VERY MINOR ethics concerns only"]

**Final Justification:**

Given the clarifications on all points and the additional experiments, I am now convinced the proposed method does have specific technical advantages.

**Limitations:**

The paper does not present any apparent risks of negative societal impact.

**Paper Formatting Concerns:**

No major formatting issues to report.

**Quality:**

2

**Strengths And Weaknesses:**

## Strengths
- The proposed GDN architecture is sound, well-motivated, and elegantly combines the strengths of GNNs and diffusion models to tackle the specific challenges of ABMs.
- The paper is clear and well written
- The ablation study that removes the GNN component clearly and effectively demonstrates the value of explicitly modeling the graph-structured interactions.
- The authors provide detailed architectural choices and experiment details, supporting reproducibility.


## Weaknesses
- The proposed framework requires full access to an existing ABM simulator to generate a ramified dataset. This restricts its use  and limits its immediate applicability to real-world applications where the simulator may be unavailable or partially known.
- The empirical comparison is restricted to a single ablation (which removes the GNN component). The paper does not compare against alternative differentiable surrogates such as Transformer-based simulators [Casert et al. 2024] or neural emulators for cellular automata [Grattarola et al.], making it harder to contextualize the contribution within the growing literature on learnable simulators.
- The diffusion-removed variant of the model is discussed but not included in the experimental evaluation. Including this variant would help isolate the contribution of the diffusion component. Moreover, in this ablation, the authors should consider comparing against temporal GNN baselines [1], which are a natural alternative for modeling sequential agent behavior. Such baselines would offer a stronger baseline for GDN.
- The framework requires generating a ramified simulation dataset, which can be expensive. This could limit applicability in large-scale ABMs with many time steps, high agent counts, or other real-time constraints.

[1] Gravina et al. Deep learning for dynamic graphs: models and benchmarks. In IEEE TNNLS 2024.

---

> ### Author Rebuttal · Authors · 2025-07-31
>
> We appreciate the reviewer’s detailed and constructive comments. We reply by numbering the weaknesses by W1, W2, … and the questions by Q1, Q2, …, grouping weaknesses and questions where relevant. For each reply, we start with a one-sentence TL;DR and then develop our argument.
>
> # W1: Need access to the ABM
> **TL;DR:** ABMs cover lots of different systems, but in principle, the method can be used on data with no existing ABM too, granted that sufficient agent/context pairs and transitions are observed in the data.
>
> Agent-based models (ABMs) are widely used to represent individual behavior and interactions across disciplines, and after more than three decades of development, it is likely that for many real-world datasets, an ABM exists that closely approximates the underlying data-generating process. The central challenge lies in aligning such ABMs with observational data.
>
> That said, our method is not limited to synthetic data. In principle, it could be applied directly to real-world data, provided that the data includes sufficient repeated instances of similar individuals facing similar interaction structures. Taking the Schelling model as an example: in our synthetic setup, we generated 500 alternative futures for an agent moving to a new location given the characteristics of its neighbors. If we had real-world data from, say, 500 households with matching characteristics and neighborhood compositions, their observed behaviors could serve as a real-world ramifications dataset.
>
> While this paper focuses on synthetic data as a necessary first step, we are actively exploring applications to real-world datasets as part of ongoing and future research.
>
> # W2, W3, Q2: Baselines
> **TL;DR:** GNN models completely fail to learn stochastic behavior, and thus lead to much lower micro- and macro performance, while alternative baselines are not general enough for ABMs.
>
> The reviewer proposes three further possible choices for ablations/baselines:
>
> 1. Perform an ablation that removes the diffusion component while keeping the GNN.
> 2. Consider alternative GNN models (e.g., Grattarola et al., temporal GNNs).
> 3. Compare to Casert et al. (2024).
>
> We tackle each of these one by one.
>
> **1. Ablation without diffusion**
>
> We conducted additional experiments using the ablated model suggested by the reviewer, which excludes the diffusion component while retaining the GNN. This modification is non-trivial, as the original surrogate relies on gradients from the diffusion model to train the GNN. In the ablated version, we instead minimize the MSE between the true and predicted next agent states. The only architectural change is in the MLP following the aggregation pass: its output dimension is set to the agent state dimension rather than the embedding dimension. While the original GDN uses hidden layers of size [32, 64, 128], the ablated model adopts a symmetrical structure: [32, 64, 128, 128, 64, 32]. We trained for 100 epochs with the Adam optimizer (learning rate 2×10⁻⁵) and mini-batch size of 16.
>
> Our results clearly show that the GNN successfully learns to predict agents’ future states when the underlying dynamics are deterministic. However, it performs poorly in the presence of stochastic rules, resulting in significantly lower overall performance compared to the GDN.
>
> The results for the predator-prey model are summarized in the tables below (mean±standard deviation over 8 runs). For each parameter, the second and third columns reproduce the results from Figure 4 (GDN and diffusion-only ablation), while the last column reports results from the GNN-only ablation as suggested by the reviewer. We observe that both macro-level and micro-level errors are substantially higher in the GNN-only ablation, and often worse than the diffusion-only model. This is expected, as stochastic rules are dominant in most parameterizations, making the GNN's deterministic forecasting approach inadequate.
>
> Predator-Prey macro-level:
>
> |Parameter |GDN |Diffusion-only |GNN-only|
> |-|-|-|-|
> |\$\Psi\_1\$ |0.079±0.030 |0.266±0.046 |0.940±0.082 |
> |\$\Psi\_2\$ |0.323±0.149 |1.204±0.135 |0.917±0.198 |
> |\$\Psi\_3\$ |0.206±0.073 |0.508±0.084 |0.744±0.235 |
> |\$\Psi\_4\$ |0.324±0.115 |0.768±0.074 |0.999±0.254 |
>
> Predator-Prey micro-level:
>
> |Parameter |GDN |Diffusion-only |GNN-only |
> |-|-|-|-|
> | \$\Psi\_1\$ |0.0091±0.0005 |0.0521±0.0024 |0.0750±0.0014|
> | \$\Psi\_2\$ |0.0013±0.0002 |0.0654±0.0102 |0.0122±0.0022|
> | \$\Psi\_3\$ |0.0096±0.0008 |0.0812±0.0028 |0.0540±0.0034|
> | \$\Psi\_4\$ |0.0041±0.0003 |0.0784±0.0030 |0.0386±0.0019|
>
> A slightly different picture emerges in the Schelling model for parameterization $\xi_1$. Here, nearly all agents remain stationary during the test period, leading to effectively deterministic dynamics. Consequently, the GNN-only surrogate achieves a micro-level EMD of 0.73±0.07, which is comparable to the GDN’s performance (0.71±0.06). However, from preliminary results (mean and std over 6 experiments, instead of 8 due to time) on parameterization $\xi_3$ , the most stochastic configuration, the GNN-only surrogate performs poorly with EMD (9.73±0.06) against GDN’s performance (2.81±0.11).
>
> In the final version of the paper we will include the full results from this additional ablation study.
>
> **2. Alternative GNN models**
>
> We argue that all GNN-based models share the same structural limitations observed in the GNN-only ablation examined above, and therefore exhibit similar performance. This class includes the approach by Grattarola et al., which in fact was applied to deterministic cellular automata, as well as the temporal GNNs discussed in the reference suggested by the reviewer  (Gravina et al. 2024)
>
> **3. Casert et al.**
>
> The reviewer suggested using the Transformer-based model by Casert et al. as a baseline. While it is the only stochastic emulator we are aware of, it is not applicable to the ABMs we consider. Their method is tailored to physical systems in continuous time and relies on explicitly enumerating allowed transitions between configurations. This is feasible when transitions involve small local changes, as in particle systems, but not in ABMs where agents can make large, unconstrained moves (e.g., in Schelling) or where the number of agents varies over time (e.g., predator-prey). In contrast, our ramifications approach samples from the space of possible configurations without requiring predefined transitions, making it suitable for ABMs but incompatible with the Casert et al. methodology.
>
> # W4, Q1: Scaling
> **TL;DR:** Our surrogate model training cost scales linearly w.r.t. time-horizon and number of ramifications and sublinearly w.r.t. number of agents. Increasing time-horizon and number of agents increases stochastic coverage, so the number of ramifications can be reduced.
>
> We acknowledge the need to discuss computational cost and scalability. Training our GDN on 9 timesteps, 500 ramifications, and ~2 000 agents (1 950 Schelling, 2 048 Predator–Prey) takes ≈ 37 s per epoch. Because each step processes one timestep and one ramification, runtime grows linearly with the number of ramifications or timesteps and only sub-linearly with the agent count (see reply to Reviewer BiVF). We use multiple stochastic branches during training to increase the statistical coverage of the ABM micro-dynamics. A new experiment on the predator–prey model (parameter set $\Psi_1$​) confirms that increasing R improves EMD and sMAPE until they match the main-paper results, while training time scales linearly.
>
> |Ramifications |EMD |sMAPE |Time/epoch |
> |-|-|-|-|
> |50 |0.0159 |0.179 |3.7 s |
> |100 |0.0138 |0.108 |7.4 s |
> |250 |0.0096 |0.088 |18.5 s |
> |500 |0.0091\* |0.079\* |37.1 s |
>
> *: Mean values from the paper micro and macro evaluations.
>
> We also agree that generating hundreds of futures for each state may be prohibitive in systems with many agents or longer time horizons. However, larger systems naturally provide more independent samples of similar conditions. In such cases, fewer ramifications may suffice to achieve similar statistical coverage. To explore the trade-off between data generation cost and predictive quality, we trained GDN on three Predator-Prey datasets (parameter $\Psi_1$) with increasing agent counts and decreasing ramifications, keeping the number of agent transitions roughly constant. All other hyperparameters remain as in the main experiments. Results are shown below.
>
> |Grid size |Agents |Ramifications |Time/epoch |sMAPE |EMD |
> |-|-|-|-|-|-|
> |64 |8192 |125 |26.94 s |0.076 |0.0121 |
> |48 |4608 |250 |32.32 s |0.033 |0.0104 |
> |32 |2048 |500 |37.09 s |0.079\* |0.0091\* |
>
> *: Mean values from the paper micro and macro evaluations.
>
> We find that sMAPE remains close to the original results (and is notably lower for the 48-grid case), while EMD increases slightly but stays near the reported average. We will include these scaling results and the discussion above in the revised manuscript and supplement to better communicate the computational footprint and scalability of our framework.
>
> # Q3: GNN details
>
> We use a simple MessagePassing GNN, where each agent’s state is passed as the message, aggregated, and fed into an MLP. As the reviewer has correctly pointed out, the choice of the MessagePassing aggregation scheme can impact expressiveness and inductive bias. The most commonly used aggregations for this setting are sum, mean and max. In our experiments we used “sum” for Predator-Prey and “mean” for Schelling since there is dependency on the degree in its dynamics. Minimal experimental evaluations allow to direct the practitioner towards the one which seems promising for the task at hand.
> In the future we would like to explore ‘aggregation-agnostic’ approaches, in order to reduce inductive bias. Promising options are relying on Attention weighting or leveraging PowerMeanAggregation or MultiAggregation operators from Pytorch.
>
> ---
> We welcome any follow-up questions and would be happy to provide further details or clarifications.

---

> > ### Comment · Reviewer_McsJ · 2025-08-03
> >
> > Thank you for your detailed reply. I appreciate the clarifications on all points and the additional experiments. The method does have specific technical advantages.

---

### Official Review · Reviewer_BiVF · 2025-06-30

**Clarity:** 3
**Significance:** 2
**Originality:** 3
**Rating:** 4
**Confidence:** 3

**Summary:**

This paper addresses an important yet challenge problem making Agent-Based Models (ABMs) differentiable, particular by making the behavior of each agent differentiable. The authors propose Graph Diffusion Networks (GDNs), a novel combination of GNN and diffusion models to learn differentiable ABM surrogates. The GNN capture the interactions of agents, and the diffusion model for stochasticity behavior. The method is evaluated on the Schelling segregation model and a Predator-Prey ecosystem, showing strong fidelity in reproducing both micro-level (individual) and macro-level (aggregate) behaviors.

**Questions:**

See weaknesses point 2 for why not compare using other existing methods.

**Ethical Concerns:**

["NO or VERY MINOR ethics concerns only"]

**Final Justification:**

I appreciated the feedback from the author, I will keep my original rating

**Limitations:**

The author discussed several key limitations. See weakness point 1 for limitation of scalability

**Quality:**

3

**Strengths And Weaknesses:**

The paper is very well written and presents a clear and comprehensive review of related work. It is well structured, with a strong motivation and background. The proposed method is novel and demonstrates strong experimental performance compared to the baseline. The limitations of the current approach are also thoughtfully discussed.

### weaknesses:

(1) The paper does not clearly state how large the ABMs used in the experiments are — there's no mention of the number of agents, grid size, or how dense the interactions are. This information is missing from the main text and, if only available in the supplement, should really be included in the main paper. From the figures and setup, it seems these are small-scale ABMs with less than 500 agents.
That’s important, because the proposed method involves several computationally intensive components: diffusion models with many sequential steps, a GNN applied per agent, and a ramification process that generates hundreds of alternate futures per timestep. These choices likely make scaling to larger systems quite costly.
Since the primary motivation for making agent behavior differentiable is to enable learning and data fitting in complex, realistic ABMs, a method that scales only to small systems may fall short of this goal.  A discussion of computational efficiency is needed to understand where and how this method could realistically be used.

(2) While the authors focus on individual-level modeling, both ABMs used (Schelling and Predator–Prey) have well-understood emergent dynamics that could also be modeled by existing surrogate approaches. The decision not to compare against other surrogate models -- as discussed in the background section -- is not fully justified. Even if these baselines do not model per-agent transitions, they are appropriate for comparison on the macro-level forecasting tasks Including such baselines would strengthen the empirical claims and clarify the added value of modeling individual behavior explicitly.

---

> ### Author Rebuttal · Authors · 2025-07-31
>
> We thank the reviewer for the thoughtful and constructive feedback. We reply by numbering the weaknesses by W1, W2, … and the questions by Q1, Q2, …, grouping weaknesses and questions where relevant. For each reply, we start with a one-sentence TL;DR and then develop our argument.
>
> ---
> # W1: Scaling
> **TL;DR**: Our surrogate model has been tested on mid-sized ABMs, the training time scales linearly with number of ramifications and timesteps, and sublinearly with number of agents.
>
> We agree that the main manuscript should state the concrete dimensions of the testbeds and clarify how the method scales.  We will move the numbers that are currently in the supplement to the main text (§4) and add the following discussion.
>
> | Component           | Schelling | Predator‑Prey |
> | ------------------- | --------- | ------------- |
> | Grid size           | 51 x 51   | 32 x 32       |
> | Initial density     | 0.75      | 0.30          |
> | Number of agents    | 1,950     | 2,048         |
> | Colour/Kind ratio | 1:1     | 1:1         |
>
>
> These settings are in the range of mid-sized ABMs: each training epoch processes about $\sim 9\times10^{6}$ agent transitions (9 time steps x 2000 agents x 500 ramified futures).  On a single RTX-A6000, one epoch takes ≈ 37 seconds.
>
> The reviewer is correct that an ABM containing millions of agents can stress any learning algorithm. As we have mentioned in the discussion with Reviewer McsJ, the training time of the model grows linearly with the number of ramifications provided to the network during training, as well as the number of timesteps. On the other hand, increasing the number of agents in the system (for example by increasing the grid size and keeping the density of agents constant) increases the training time sublinearly. In fact, with mid-sized systems, we can train the network with batches that correspond to the whole set of agents. Thus, the number of training iterations per epoch does not change, but rather the size of the batch in input to the network. Considering the Predator-Prey model, if we increase the grid size from 32 to 64, going from 2048 agents to 8192 (4 times as many agents), we have an increase from 37.09 s/epoch to 108.37 s/epoch, which is 2.92 times larger. Similarly, by increasing from grid size 32 to 48, we increase by a factor of 2.25 the number of agents, but the training time increases by a factor of 1.72. Regarding inference, the computational cost to produce the next agent state is always the same. However, it will take longer for larger systems to simulate the evolution of the entire system. Below we show a table with the training time and inference time for the 3 settings of grid sizes we have discussed, where inference time refers to the time necessary to generate 1 timestep of a simulation of N. agents with the trained GDN. All other hyper-parameters are the same as in the paper experiments. We include the evaluation metrics at the macro and micro level used in the paper, to assess that even with a larger system the dynamics are learned by our methodology.
>
> | Grid size |  N. agents | Time/epoch |Inference time |  sMAPE   |    EMD    |
> |-------------|---------------|---------|--------------|-----------|-------------|
> |      64     |      8192     | 108 s |  1.6 s/it. |0.041  |  0.0080 |
> |      48     |      4608     | 64 s   |  0.7 s/it. |0.044  | 0.0084  |
> |      32     |      2048     |  37 s  |  0.3 s/it.|0.079*|  0.0091*|
>
> *: Mean values from the paper micro and macro evaluations.
>
> Training the model with a ramified dataset of R ramifications is a choice made to allow the model to be exposed to many agent transitions ($\sim 9\times10^{6}$ in our experiments). Increasing the number of agents in the system increases the number of agent transitions available in the dataset, expanding the statistical coverage provided to the model to learn. In our previous experiment, the models with higher numbers of agents performed better both at the macro and micro level than in the original experiments. The number of ramifications R could be decreased to potentially get the same result quality of the paper experiments. We ran an experiment by adjusting the number of ramifications R according to the number of agents so that we have a similar number of agent transitions in the training set as in the original experiments. Given the time constraints of the rebuttal period, we were able to train one model for each grid size configuration on one dataset of Predator-Prey with parameter $\Psi_1$ and evaluate the model as done in the paper. The results are shown in the table below. We find that the sMAPE is around the value found in the original experiments (actually lower for grid size 48), and the mean EMD is slightly higher, but close to the mean value from the original experiments. We will add the clarifying discussion above and additional scaling experiments with more hyperparameter values to the revised manuscript and supplement material.
>
>
> | Grid size |  N. agents |  N. ramifications  |  Time/epoch  |    sMAPE   |    EMD    |
> |-----|----|-----|--------|----|----|
> |      64     |      8192     |    125     |   27 s |     0.076     |  0.0121   |
> |      48     |      4608     |     250      |   32 s |     0.033     |  0.0104   |
> |      32     |      2048     |     500      |   37 s |     0.079*    |  0.0091* |
>
> *: Mean values from the paper micro and macro evaluations.
>
> Inference time refers to the time needed to produce 1 timestep of a simulation of N agents with the trained GDN.
>
> Regarding scaling to even larger systems (e.g. in the range of $10^4$ to $10^6$), it would be more appropriate to train the model on batches made of subgraphs of the system configuration, as it would become troublesome to optimize the model on batches of tens of thousands or millions of samples. However, apart from batch optimization, the methodology would remain unchanged.
>
> ---
> # W2, Q1: Macro-level baselines
> **TL;DR**: While surrogate models for aggregate dynamics do exist, they struggle to reproduce trends that are absent from the training data. In contrast, our approach (based on learning individual behavior) can generalize to these unseen dynamics.
> Most existing surrogate approaches do not actually learn system dynamics. Instead, quoting from our manuscript, surrogate models typically learn directly the mapping from parameters $\Theta$ to static summary statistics, disregarding individual behavior and model dynamics.
>
> There are a few exceptions. For example, Fadikar et al. [1] use functional Gaussian Processes to capture the shape of epidemic curves, while Kieu et al. [2] employ LSTMs to forecast time series of pedestrian counts. However, these approaches share two key limitations: (i) they model aggregate variables rather than micro-level behavior, and (ii) they rely on statistical models fitted to in-sample data.
>
> As a result, these models perform well when out-of-sample dynamics resemble those seen during training, but their performance deteriorates when the dynamics change. For instance, Fadikar et al. focus on cumulative case counts in epidemics, which follow a monotonic trajectory. In such cases, using early time steps to predict future ones is a reasonable strategy. But this assumption breaks down in more complex settings.
>
> Consider the predator-prey model shown in Figure 3 (top panels), where both predator and prey populations grow and then decline. These dynamics are far from monotonic. A surrogate trained on aggregate counts during the early phase (when populations are rising) is unlikely to capture the subsequent decline.
>
> To illustrate this point, we implemented a simple AR(1) process as a representative of aggregate time-series-based models. We trained it on just 10 time steps (matching the surrogate’s training window). Given this limited horizon, using more complex models like LSTMs would be inappropriate.
>
> The AR(1) surrogate substantially underperforms our individual-level surrogate. For example, in scenario $\Psi_1$—characterized by oscillatory predator-prey dynamics—the AR(1) either predicts indefinite growth or convergence to a steady state. It fails to capture the true cyclical behavior, resulting in a significantly higher sMAPE. Even in the simpler scenario $\Psi_2$, where dynamics are monotonic, the AR(1) model incorrectly predicts convergence to a steady state (e.g., a predator and prey population of 60), again leading to large errors.
>
> The table below reports the sMAPE (mean and standard deviation) across four scenarios. As shown, the AR(1) surrogate performs poorly for all scenarios, with errors far exceeding those of our micro-founded surrogate reported in Figure 4.
>
> | Scenario | sMAPE (mean) | sMAPE (std) |
> |---|---|----|
> | $\Psi_1$ | 0.82 | 0.08 |
> | $\Psi_2$ | 0.82 | 0.06 |
> | $\Psi_3$ | 0.44 | 0.06 |
> | $\Psi_4$ | 0.96  | 0.12|
>
> In summary, we deliberately chose not to include comparisons with macro-level dynamic surrogates in the main paper because their limitations are structural: they rely entirely on aggregate time series observed during training. As such, they are fundamentally incapable of extrapolating to qualitatively different out-of-sample dynamics—precisely the type of generalization our setting requires. The AR(1) results shown here support this point, illustrating that even the simplest non-monotonic patterns are missed. For this reason, we maintain that comparing our surrogate to these macro-level approaches would not provide a meaningful benchmark.
> We will include this discussion and the results on the AR(1) in the revised paper.
>
> [1] Fadikar, A., Higdon, D., Chen, J., Lewis, B., Venkatramanan, S., & Marathe, M. (2018). Calibrating a stochastic, agent-based model using quantile-based emulation. SIAM/ASA Journal on Uncertainty Quantification, 6(4), 1685-1706.
>
> [2] Kieu, M., Nguyen, H., Ward, J. A., & Malleson, N. (2024). Towards real-time predictions using emulators of agent-based models. Journal of Simulation, 18(1), 29-46.

---

> > ### Comment · Reviewer_BiVF · 2025-08-05
> >
> > Thank you for your detailed clarification. The responses addressed my questions.

---

### Note · Authors · 2025-08-13

We are pleased that all reviewers indicated their concerns were addressed during the rebuttal period, and thank them for their feedback that helps us strengthen the paper considerably.

Specifically, we will incorporate new experimental evidence in two key areas:

 1- Comparisons to alternative approaches, including a new ablation study. This new data shows that our decision not to compare against other baselines is well-justified: all plausible alternatives are either too narrow in scope or significantly underperform our method. For example, GNN-only baselines cannot capture the stochastic dynamics, while macro-level (i.e. non-individual) baselines fail to reproduce the non-monotonic behaviors absent from the training data in a time-based train-test split.

2- Scalability analysis. We clarified that our methodology has been applied to mid-sized ABMs, reported computational costs, and analyzed how performance scales with agent count and ramifications. This offers more transparent information on the scalability  of our method, and concrete guidance for applying our method to larger systems.

We believe these additions substantially improve the clarity, rigor, and breadth of the work. Beyond its immediate application to ABMs, our methodology represents a novel integration of diffusion models and GNNs that addresses the challenge of modeling decentralized, stochastic systems. This combination opens promising opportunities for advancing differentiable, neural, learnable simulators in complex systems in general, reinforcing the broader impact and significance of this work.

---

### Decision · Program_Chairs · 2025-09-17

**Decision:**

Accept (poster)

**Comment:**

This paper proposes a novel framework, which combines graph neural networks and diffusion models to learn a fully differentiable surrogate model directly from ABM-generated simulation data. The reviewers pointed out that the paper is clear and well-written and the proposed architecture is sound. The proposed method also shows strong performance compared with the baselines.

The reviewers also pointed out some concerns, including missing details, scalability, and a lack of discussion on related works, which are well addressed by the authors during the rebuttal. I encourage authors to include all the rebuttal contents in their camera-ready version.